# OPAL: Offline Primitive Discovery for Accelerating Offline Reinforcement Learning

**Anurag Ajay**[*1]**, Aviral Kumar**[3]**, Pulkit Agrawal**[1]**, Sergey Levine**[2,3]**, Ofir Nachum**[2]
[1]MIT,   [2]Google Research,   [3]UC Berkeley

## Abstract

Reinforcement learning (RL) has achieved impressive performance in a variety of *online* settings in which an agent's ability to query the environment for transitions and rewards is effectively unlimited. However, in many practical applications, the situation is reversed: an agent may have access to large amounts of undirected offline experience data, while access to the online environment is severely limited. In this work, we focus on this offline setting. Our main insight is that, when presented with offline data composed of a variety of behaviors, an effective way to leverage this data is to extract a continuous space of recurring and temporally extended *primitive* behaviors before using these primitives for downstream task learning. Primitives extracted in this way serve two purposes: they delineate the behaviors that are supported by the data from those that are not, making them useful for avoiding distributional shift in offline RL; and they provide a degree of temporal abstraction, which reduces the effective horizon yielding better learning in theory, and improved offline RL in practice. In addition to benefiting offline policy optimization, we show that performing offline primitive learning in this way can also be leveraged for improving few-shot imitation learning as well as exploration and transfer in online RL on a variety of benchmark domains. Visualizations and code are available at https://sites.google.com/view/opal-iclr

## 1 Introduction

Reinforcement Learning (RL) systems have achieved impressive performance in a variety of *online* settings such as games (Silver et al., 2016; Tesauro, 1995; Brown & Sandholm, 2019) and robotics (Levine et al., 2016; Dasari et al., 2019; Peters et al., 2010; Parmas et al., 2019; Pinto & Gupta, 2016; Nachum et al., 2019a), where the agent can act in the environment and sample as many transitions and rewards as needed. However, in many practical applications the agent's ability to continuously act in the environment may be severely limited due to practical concerns (Dulac-Arnold et al., 2019). For example, a robot learning through trial and error in the real world requires costly human supervision, safety checks, and resets (Atkeson et al., 2015), rendering many standard online RL algorithms inapplicable (Matsushima et al., 2020). However, in such settings we might instead have access to large amounts of previously logged data, which could be logged from a baseline hand-engineered policy or even from other related tasks. For example, in self-driving applications, one may have access to large amounts of human driving behavior; in robotic applications, one might have data of either humans or robots performing similar tasks. While these offline datasets are often undirected (generic human driving data on various routes in various cities may not be directly relevant to navigation of a specific route within a specific city) and unlabelled (generic human driving data is often not labelled with the human's intended route or destination), this data is still useful in that it can inform the algorithm about what is *possible* to do in the real world, without the need for active exploration.

In this paper, we study how, in this offline setting, an effective strategy to leveraging unlabeled and undirected past data is to utilize unsupervised learning to extract potentially useful and temporally extended *primitive* skills to learn what types of behaviors are *possible*. For example, consider a dataset of an agent performing undirected navigation in a maze environment (Figure 1). While the dataset does not provide demonstrations of exclusively one specific point-to-point navigation task,

---

*Work done during an internship at Google Brain

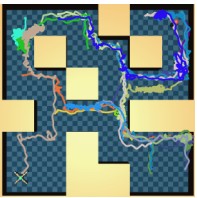 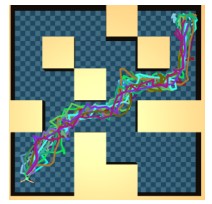 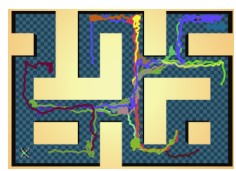 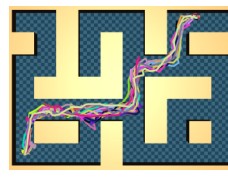

(a) medium (diverse)  (b) medium (CQL+OPAL)  (c) large (diverse)  (d) large (CQL+OPAL)

Figure 1: Visualization of (a subset of) *diverse* datasets for (a) antmaze medium and (c) antmaze large, along with trajectories sampled from **CQL+OPAL** trained on *diverse* datasets of (b) antmaze medium and (d) antmaze large.

it nevertheless presents clear indications of which temporally extended behaviors are useful and natural in this environment (e.g., moving forward, left, right, and backward), and our unsupervised learning objective aims to distill these behaviors into temporally extended primitives. Once these locomotive primitive behaviors are extracted, we can use them as a compact constrained temporally-extended action space for learning a task policy with offline RL, which only needs to focus on task relevant navigation, thereby making task learning easier. For example, once a specific point-to-point navigation is commanded, the agent can leverage the learned primitives for locomotion and only focus on the task of navigation, as opposed to learning locomotion and navigation from scratch.

We refer to our proposed unsupervised learning method as *Offline Primitives for Accelerating offline reinforcement Learning* (OPAL), and apply this basic paradigm to offline RL, where the agent is given a single offline dataset to use for both the initial unsupervised learning phase and then a subsequent task-directed offline policy optimization phase. Despite the fact that no additional data is used, we find that our proposed unsupervised learning technique can dramatically improve offline policy optimization compared to performing offline policy optimization on the raw dataset directly. To the best of our knowledge, ours is the first work to theoretically justify and experimentally verify the benefits of primitive learning in offline RL settings, showing that hierarchies can provide temporal abstraction that allows us to reduce the effect of compounding errors issue in offline RL. These theoretical and empirical results are notably in contrast to previous related work in *online* hierarchical RL (Nachum et al., 2019b), which found that improved exploration is the main benefit afforded by hierarchically learned primitives. We instead show significant benefits in the *offline* RL setting, where exploration is irrelevant.

Beyond offline RL, and although this isn't the main focus of the work, we also show the applicability of our method for accelerating RL by incorporating OPAL as a preprocessing step to standard online RL, few-shot imitation learning, and multi-task transfer learning. In all settings, we demonstrate that the use of OPAL can improve the speed and quality of downstream task learning.

## 2 RELATED WORK

**Offline RL.** Offline RL presents the problem of learning a policy from a fixed prior dataset of transitions and rewards. Recent works in offline RL (Kumar et al., 2019; Levine et al., 2020; Wu et al., 2019; Ghasemipour et al., 2020; Jaques et al., 2019; Fujimoto et al., 2018) constrain the policy to be close to the data distribution to avoid the use of out-of-distribution actions (Kumar et al., 2019; Levine et al., 2020). To constrain the policy, some methods use distributional penalties, as measured by KL divergence (Levine et al., 2020; Jaques et al., 2019), MMD (Kumar et al., 2019), or Wasserstein distance (Wu et al., 2019). Other methods first sample actions from the behavior policy and then either clip the maximum deviation from those actions (Fujimoto et al., 2018) or just use those actions (Ghasemipour et al., 2020) during the value backup to stay within the support of the offline data. In contrast to these works, OPAL uses an offline dataset for unsupervised learning of a continuous space of primitives. The use of these primitives for downstream tasks implicitly constrains a learned primitive-directing policy to stay close to the offline data distribution. As we demonstrate in our experiments, the use of OPAL in conjunction with an off-the-shelf offline RL algorithm in this way can yield significant improvement compared to applying offline RL to the dataset directly.

**Online skill discovery.** There are a number of recent works (Eysenbach et al., 2018; Nachum et al., 2018a; Sharma et al., 2019) which use unsupervised objectives to discover skills and use the discovered skills for planning (Sharma et al., 2019), few-shot imitation learning, or online RL (Eysenbach et al., 2018; Nachum et al., 2018a). However, these works focus on online settings and assume access to the environment. In contrast, OPAL focuses on settings where a large dataset of diverse behaviors is provided but access to the environment is restricted. It leverages these static offline datasets to discover primitive skills with better state coverage and avoids the exploration issue of learning primitives from scratch.

**Hierarchical policy learning.** Hierarchical policy learning involves learning a hierarchy of policies where a low-level policy acts as primitive skills and a high-level policy directs the low-level policy to solve a task. While some works (Bacon et al., 2017; Stolle & Precup, 2002; Peng et al., 2019) learn a discrete set of lower-level policies, each behaving as a primitive skill, other works (Vezhnevets et al., 2017; Nachum et al., 2018b; 2019a; Hausman et al., 2018) learn a continuous space of primitive skills representing the lower-level policy. These methods have mostly been applied in online settings. However, there have been some recent variants of the above works (Lynch et al., 2020; Shankar & Gupta, 2020; Krishnan et al., 2017; Merel et al., 2018) which extract skills from a prior dataset and using it for either performing tasks directly (Lynch et al., 2020) or learning downstream tasks (Shankar & Gupta, 2020; Krishnan et al., 2017; Merel et al., 2018) with online RL. While OPAL is related to these works, we mainly focus on leveraging the learned primitives for asymptotically improving the performance of offline RL; i.e., both the primitive learning and the downstream task must be solved using a single static dataset. Furthermore, we provide performance bounds for OPAL and enumerate the specific properties an offline dataset should possess to guarantee improved downstream task learning, while such theoretical guarantees are largely absent from existing work.

## 3 PRELIMINARIES

We consider the standard Markov decision process (MDP) setting (Puterman, 1994), specified by a tuple $\mathcal{M} = (\mathcal{S}, \mathcal{A}, \mathcal{P}, \mu, r, \gamma)$ where $\mathcal{S}$ represents the state space, $\mathcal{A}$ represents the action space, $\mathcal{P}(s'|s, a)$ represents the transition probability, $\mu(s)$ represents the initial state distribution, $r(s, a) \in (-\mathrm{R}_{\max}, \mathrm{R}_{\max})$ represents the reward function, and $\gamma \in (0, 1)$ represents the discount factor. A policy $\pi$ in this MDP corresponds to a function $\mathcal{S} \rightarrow \Delta(\mathcal{A})$, where $\Delta(\mathcal{A})$ is the simplex over $\mathcal{A}$. It induces a discounted future state distribution $d^\pi$, defined by $d^\pi(s) = (1 - \gamma) \sum_{t=0}^{\infty} \gamma^t \mathcal{P}(s_t = s|\pi)$, where $\mathcal{P}(s_t = s|\pi)$ is the probability of reaching the state $s$ at time $t$ by running $\pi$ on $\mathcal{M}$. For a positive integer $k$, we use $d_k^\pi(s) = (1 - \gamma^k) \sum_{t=0}^{\infty} \gamma^{tk} \mathcal{P}(s_{tk} = s|\pi)$ to denote the every-$k$-step state distribution of $\pi$. The return of policy $\pi$ in MDP $\mathcal{M}$ is defined as $J_{\mathrm{RL}}(\pi, \mathcal{M}) = \frac{1}{1-\gamma} \mathbb{E}_{s \sim d^\pi, a \sim \pi(a|s)}[r(s, a)]$. We represent the reward- and discount-agnostic environment as a tuple $\mathcal{E} = (\mathcal{S}, \mathcal{A}, \mathcal{P}, \mu)$.

We aim to use a large, unlabeled, and undirected experience dataset $\mathcal{D} := \{\tau_i^r := (s_t, a_t)_{t=0}^{c-1}\}_{i=1}^N$ associated with $\mathcal{E}$ to extract primitives and improve offline RL for downstream task learning. To account for the fact that the dataset $\mathcal{D}$ may be generated by a mixture of diverse policies starting at diverse initial states, we assume $\mathcal{D}$ is generated by first sampling a behavior policy $\pi \sim \Pi$ along with an initial state $s \sim \kappa$, where $\Pi, \kappa$ represent some (unknown) distributions over policies and states, respectively, and then running $\pi$ on $\mathcal{E}$ for $c$ time steps starting at $s_0 = s$. We define the probability of a sub-trajectory $\tau := (s_0, a_0, \ldots, s_{c-1}, a_{c-1})$ in $\mathcal{D}$ under a policy $\pi$ as $\pi(\tau) = \kappa(s_0) \prod_{t=1}^{c-1} \mathcal{P}(s_t|s_{t-1}, a_{t-1}) \prod_{t=0}^{c-1} \pi(a_t|s_t)$, and the conditional probability as $\pi(\tau|s) = 1[s = s_0] \prod_{t=1}^{c-1} \mathcal{P}(s_t|s_{t-1}, a_{t-1}) \prod_{t=0}^{c-1} \pi(a_t|s_t)$. In this work, we will show how to apply unsupervised learning techniques to $\mathcal{D}$ to extract a continuous space of primitives $\pi_\theta(a|s, z)$, where $z \in \mathcal{Z}$, the latent space inferred by unsupervised learning. We intend to use the learned $\pi_\theta(a|s, z)$ to asymptotically improve the performance of offline RL for downstream task learning. For offline RL, we assume the existence of a dataset $\mathcal{D}^r := \{\tau_i^r := (s_t, a_t, r_t)_{t=0}^{c-1}\}_{i=1}^N$, corresponding to the same sub-trajectories in $\mathcal{D}$ labelled with MDP rewards. Additionally, we can use the extracted primitives for other applications such as few-shot imitation learning, online RL, and online multi-task transfer learning. We review the additional assumptions for these applications in Appendix A.

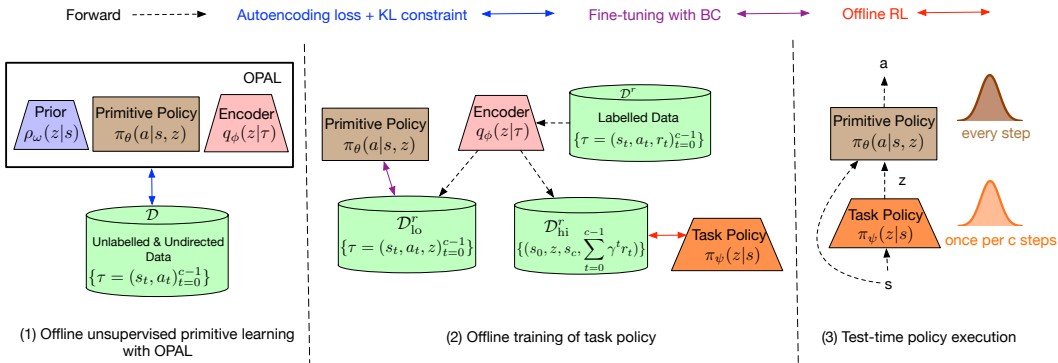

Figure 2: Overview of offline RL with OPAL. OPAL is trained on unlabelled data $\mathcal{D}$ using autoencoding objective. For offline RL, the encoder first labels the reward-labelled data $\mathcal{D}^r$ with latents, and divides it into $\mathcal{D}^r_{\text{hi}}$ and $\mathcal{D}^r_{\text{lo}}$. The task policy $\pi_\psi$ is trained on $\mathcal{D}^r_{\text{hi}}$ using offline RL while the primitive policy $\pi_\theta$ is finetuned on $\mathcal{D}^{\text{lo}}$ using behavioral cloning (BC).

## 4   OFFLINE RL WITH OPAL

In this section, we elaborate on OPAL, our proposed method for extracting primitives from $\mathcal{D}$ and then leveraging these primitives to learn downstream tasks with offline RL. We begin by describing our unsupervised objective, which distills $\mathcal{D}$ into a continuous space of latent-conditioned and temporally-extended primitive policies $\pi_\theta(a|s,z)$. For learning downstream tasks with offline RL, we first label $\mathcal{D}^r$ with appropriate latents using the OPAL encoder $q_\phi(z|\tau)$ and then learn a policy $\pi_\psi(z|s)$ which is trained to sample an appropriate primitive every $c$ steps to optimize a specific task, using any off-the-shelf offline RL algorithm. A graphical overview of offline RL with OPAL is shown in Figure 2. While we mainly focus on offline RL, we briefly discuss how to use the learned primitives for few-shot imitation learning, online RL, and multi-task online transfer learning in section 5 and provide more details in Appendix A.

### 4.1   EXTRACTING TEMPORALLY-EXTENDED PRIMITIVES FROM DATA

We would like to extract a continuous space of temporally-extended primitives $\pi_\theta(a|s,z)$ from $\mathcal{D}$ which we can later use as an action space for learning downstream tasks with offline RL. This would reduce our effective task horizon, thereby making the downstream learning easier, as well as allow the downstream policy to stay close to the offline data distribution, thereby bringing stability to the downstream learning. We propose the following objective for learning $\pi_\theta$, incorporating an auto-encoding loss function with a KL constraint to encourage better generalization:

$$\min_{\theta,\phi,\omega} J(\theta,\phi,\omega) = \hat{\mathbb{E}}_{\tau\sim\mathcal{D},z\sim q_\phi(z|\tau)}\left[-\sum_{t=0}^{c-1}\log\pi_\theta(a_t|s_t,z)\right] \tag{1}$$

$$\text{s.t. } \hat{\mathbb{E}}_{\tau\sim\mathcal{D}}[\text{D}_{\text{KL}}(q_\phi(z|\tau)||\rho_\omega(z|s_0))] \leq \epsilon_{\text{KL}} \tag{2}$$

where $\hat{\mathbb{E}}$ indicates empirical expectation. The learned components of this objective may be interpreted as encoder, decoder, and prior:

**Encoder:** $q_\phi(z|\tau)$ encodes the trajectory $\tau$ of state-action pairs into a distribution in latent space and gives out parameters of that distribution. In our case, we represent $q_\phi$ with a bidirectional GRU which takes in $\tau$ and gives out parameters of a Gaussian distribution $(\mu_z^{enc}, \sigma_z^{enc})$.

**Decoder (aka Primitive Policy):** $\pi_\theta(a|s,z)$ is the latent-conditioned policy. It maximizes the conditional log-likelihood of actions in $\tau$ given the state and the latent vector. In our implementation, we parameterize it as a feed-forward neural network which takes in current state and latent vector and gives out parameters of a Gaussian distribution for the action $(\mu_a, \sigma_a)$.

**Prior/Primitive Predictor:** $\rho_\omega(z|s_0)$ tries to predict the encoded distribution of the sub-trajectory $\tau$ from its initial state. Our implementation uses a feed-forward neural network which takes in the initial state and gives out parameters of a Gaussian distribution $(\mu_z^{pr}, \sigma_z^{pr})$.

**KL-constraint (Equation 2)**. As an additional component of the algorithm, we enforce consistency in the latent variables predicted by the encoder $q_\phi(z|\tau)$ and the prior $\rho_\omega(z|s_0)$. Since our goal is to obtain a primitive $z$ that captures a temporal sequence of actions for a given sub-trajectory $\tau = (s_0, a_0, \cdots, s_{c-1}, a_{c-1})$ (as defined in Section 3), we utilize a regularization that enforces the distribution, $q_\phi(z|\tau)$ to be close to just predicting the primitive or the latent variable $z$ given the start state of this sub-trajectory, $s_0$ (i.e. $\rho_\omega(z|s_0)$). This conditioning on the initial state regularizes the distribution $q_\phi(z|\tau)$ to not overfit to the the complete sub-trajectory $\tau$ as the same $z$ should also be predictable only given $s_0$. The above form of KL constraint is inspired from past works (Lynch et al., 2020; Kumar et al., 2020a). In particular Lynch et al. (2020) add a KL-constraint (Equation 2, "Plan prior matching" in Lynch et al. (2020)) that constrains the distribution over latent variables computed only given the initial state and the goal state to the distribution over latent variables computed using the entire trajectory. Our form in Equation 2 is similar to this prior except that we do not operate in a goal-conditioned RL setting and hence only condition $\rho_\omega$ on the initial state $s_0$.

In practice, rather than solving the constrained optimization directly, we implement the KL constraint as a penalty, weighted by an appropriately chosen coefficient $\beta$. Thus, one may interpret our unsupervised objective as using a sequential $\beta$-VAE (Higgins et al., 2016). However, as mentioned above, our prior is conditioned on $s_0$ and learned as part of the optimization because the set of primitives active in $\mathcal{D}$ depends on $s_0$. If $\beta = 1$, OPAL is equivalent to a conditional VAE maximizing log probability of $\tau$ conditioned on its initial state $s_0$; see Appendix D for more details. Despite the similarities between our proposed objective and VAEs, our presentation of OPAL as a constrained auto-encoding objective is deliberate. As we will show in Section 4.3, our theoretical guarantees depend on a well-optimized auto-encoding loss to provide benefits of using learned primitives $\pi_\theta$ for downstream tasks. In contrast, a VAE loss, which simply maximizes the likelihood of observed data, may not necessarily provide a benefit for downstream tasks. For example, if the data can be generated by a single stationary policy, a VAE-optimal policy $\pi_\theta$ can simply ignore the latent $z$, thus producing a degenerate space of primitives. In contrast, when the KL constraint in our objective is weak (i.e., $\epsilon_{\mathrm{KL}} \gg 0$ or $\beta < 1$), the auto-encoding loss is encouraged to find a unique $z$ for distinct $\tau$ to optimize reconstruction loss.

## 4.2 OFFLINE RL WITH PRIMITIVES FOR DOWNSTREAM TASKS

After distilling learned primitives from $\mathcal{D}$ in terms of an encoder $q_\phi(z|\tau)$, a latent primitive policy (or decoder) $\pi_\theta(a|s, z)$, and a prior $\rho_\omega(z|s_0)$, OPAL then applies these learned models to improve offline RL for downstream tasks.

As shown in Figure 2, our goal is to use a dataset with reward labeled sub-trajectories $\mathcal{D}^r = \{\tau_i := (s_t^i, a_t^i, r_t^i)_{t=0}^{c-1}\}_{i=1}^N$ to learn a behavior policy $\pi$ that maximizes cumulative reward. With OPAL, we use the learned primitives $\pi_\theta(a|s, z)$ as low-level controllers, and then learn a high-level controller $\pi_\psi(z|s)$. To do so, we relabel the dataset $\mathcal{D}^r$ in terms of temporally extended transitions using the learned encoder $q_\phi(z|\tau)$. Specifically, we create a dataset $\mathcal{D}_{\mathrm{hi}}^r = \{(s_0^i, z_i, \sum_{t=0}^{c-1} \gamma^t r_t^i, s_c^i)\}_{i=1}^N$, where $z_i \sim q_\phi(\cdot|\tau_i)$. Given $\mathcal{D}_{\mathrm{hi}}^r$, any off-the-shelf offline RL algorithm can be used to learn $\pi_\psi(z|s)$ (in our experiments we use CQL (Kumar et al., 2020b)). As a way to ensure that the $c$-step transitions $\tau_i := (s_t^i, a_t^i, r_t^i)_{t=0}^{c-1}$ remain consistent with the labelled latent action $z_i$, we finetune $\pi_\theta(a|s, z)$ on $\mathcal{D}_{\mathrm{lo}}^r = \{((s_t^i, a_t^i)_{t=0}^{c-1}, z_i)\}_{i=1}^N$ with a simple latent-conditioned behavioral cloning loss:

$$\min_\theta \hat{\mathbb{E}}_{(\tau,z)\sim\mathcal{D}_{\mathrm{lo}}^r} \left[ -\sum_{t=0}^{c-1} \log \pi_\theta(a_t|s_t, z) \right]. \tag{3}$$

## 4.3 SUBOPTIMALITY AND PERFORMANCE BOUNDS FOR OPAL

Now, we will analyze OPAL and derive performance bounds for it in the context of offline RL, formally examining the benefit of the temporal abstraction afforded by OPAL as well as studying what properties $\mathcal{D}$ should possess so that OPAL can improve downstream task performance.

As explained above, when applying OPAL to offline RL, we first learn the primitives $\pi_\theta(a|s, z)$ using $\mathcal{D}$, and then learn a high-level task policy $\pi_\psi(z|s)$ in the space of the primitives. Let $\pi_{\psi^*}(z|s)$ be the optimal task policy. Thus the low-level $\pi_\theta$ and high-level $\pi_{\psi^*}$ together comprise a hierarchical policy, which we denote as $\pi_{\theta,\psi^*}$. To quantify the performance of policies obtained from OPAL,

we define the notion of suboptimality of the learned primitives $\pi_\theta(a|s, z)$ in an MDP $\mathcal{M}$ with an associated optimal policy $\pi^*$ as

$$\text{SubOpt}(\theta) := |J_{\text{RL}}(\pi^*, \mathcal{M}) - J_{\text{RL}}(\pi_{\theta,\psi^*}, \mathcal{M})|. \tag{4}$$

To relate $\text{SubOpt}(\theta)$ with some notion of *divergence* between $\pi^*$ and $\pi_{\theta,\psi^*}$, we introduce the following performance difference lemma.

**Lemma 4.0.1.** *If $\pi_1$ and $\pi_2$ are two policies in $\mathcal{M}$, then*

$$|J_{\text{RL}}(\pi_1, \mathcal{M}) - J_{\text{RL}}(\pi_2, \mathcal{M})| \leq \frac{2}{(1-\gamma^c)(1-\gamma)} \text{R}_{\max} \mathbb{E}_{s \sim d_c^{\pi_1}} [\text{D}_{\text{TV}}(\pi_1(\tau|s)||\pi_2(\tau|s))], \tag{5}$$

*where $\text{D}_{\text{TV}}(\pi_1(\tau|s)||\pi_2(\tau|s))$ denotes the TV divergence over $c$-length sub-trajectories $\tau$ sampled from $\pi_1$ vs. $\pi_2$ (see section 3). Furthermore,*

$$\text{SubOpt}(\theta) \leq \frac{2}{(1-\gamma^c)(1-\gamma)} \text{R}_{\max} \mathbb{E}_{s \sim d_c^{\pi^*}} [\text{D}_{\text{TV}}(\pi^*(\tau|s)||\pi_{\theta,\psi^*}(\tau|s))]. \tag{6}$$

The proof of the above lemma and all the following results are provided in Appendix B.1.

Through above lemma, we showed that the suboptimality of the learned primitives can be bounded by the total variation divergence between the optimal policy $\pi^*$ in $\mathcal{M}$ and the optimal policy acting through the learned primitives $\pi_{\theta,\psi^*}$. We now continue to bound the divergence between $\pi^*$ and $\pi_{\theta,\psi^*}$ in terms of how representative $\mathcal{D}$ is of $\pi^*$ and how optimal the primitives $\pi_\theta$ are with respect to the auto-encoding objective (equation 1). We begin with a definition of how often an arbitrary policy appears in $\Pi$, the distribution generating $\mathcal{D}$:

**Definition 1.** *We say a policy $\bar\pi$ in $\mathcal{M}$ is $\zeta$-common in $\Pi$ if $\mathbb{E}_{\pi \sim \Pi, s \sim \kappa}[\text{D}_{\text{TV}}(\pi(\tau|s)||\bar\pi(\tau|s))] \leq \zeta$.*

**Theorem 4.1.** *Let $\theta, \phi, \omega$ be the outputs of solving equation 1, such that $J(\theta, \phi, \omega) = \epsilon_c$. Then, with high probability $1 - \delta$, for any $\bar\pi$ that is $\zeta$-common in $\Pi$, there exists a distribution $H$ over $z$ such that for $\pi_\theta^H(\tau|s) := \mathbb{E}_{z \sim H}[\pi_\theta(\tau|z, s)]$,*

$$\mathbb{E}_{s \sim \kappa}[\text{D}_{\text{TV}}(\bar\pi(\tau|s)||\pi_\theta^H(\tau|s))] \leq \zeta + \sqrt{\frac{1}{2}\left(\epsilon_c + \sqrt{\frac{S_J}{\delta}} + \mathcal{H}_c\right)} \tag{7}$$

*where $\mathcal{H}_c = \mathbb{E}_{\pi \sim \Pi, \tau \sim \pi, s_0 \sim \kappa}[\sum_{t=0}^{c-1} \log \pi(a_t|s_t)]$ (i.e. a constant and property of $\mathcal{D}$) and $S_J$ is a positive constant incurred due to sampling error in $J(\theta, \phi, \omega)$ and depends on concentration properties of $\pi_\theta(a|s, z)$ and $q_\phi(z|\tau)$.*

**Corollary 4.1.1.** *If the optimal policy $\pi^*$ of $\mathcal{M}$ is $\zeta$-common in $\Pi$, and $\left\|\frac{d_c^{\pi^*}}{\kappa}\right\|_\infty \leq \xi$, then, with high probability $1 - \delta$,*

$$\text{SubOpt}(\theta) \leq \frac{2\xi}{(1-\gamma^c)(1-\gamma)} \text{R}_{\max}\left(\zeta + \sqrt{\frac{1}{2}\left(\epsilon_c + \sqrt{\frac{S_J}{\delta}} + \mathcal{H}_c\right)}\right). \tag{8}$$

As we can see, $\text{SubOpt}(\theta)$ will reduce as $\mathcal{D}$ gets *closer* to $\pi^*$ (i.e. $\zeta$ approaches 0) and better primitives are learned (i.e. $\epsilon_c$ decreases). While it might be tempting to increase $c$ (i.e. the length of sub-trajectories) to reduce the suboptimality, a larger $c$ will inevitably make it practically harder to control the autoencoding loss $\epsilon_c$, thereby leading to an increase in overall suboptimality and inducing a trade-off in determining the best value of $c$. In our experiments we treat $c$ as a hyperparameter and set it to $c = 10$, although more sophisticated ways to determine $c$ can be an interesting avenue for future work.

Till now, we have argued that there *exists* some near-optimal task policy $\pi_{\psi^*}$ if $\theta$ is sufficiently learned and $\pi^*$ is sufficiently well-represented in $\mathcal{D}$. Now, we will show how primitive learning can *improve* downstream learning, by considering the benefits of using OPAL with offline RL. Building on the policy performance analysis from Kumar et al. (2020b), we now present theoretical results bounding the performance of the policy obtained when offline RL is performed with OPAL.

| Environment | BC | BEAR | EMAQ | CQL | CQL+OPAL (ours) |
|---|---|---|---|---|---|
| antmaze medium (diverse) | 0.0 | 8.0 | 0.0 | $53.7 \pm 6.1$ | $\mathbf{81.1 \pm 3.1}$ |
| antmaze large (diverse) | 0.0 | 0.0 | 0.0 | $14.9 \pm 3.2$ | $\mathbf{70.3 \pm 2.9}$ |
| kitchen mixed | 47.5 | 47.2 | $\mathbf{70.8 \pm 2.3}$ | $52.4 \pm 2.5$ | $\mathbf{69.3 \pm 2.7}$ |
| kitchen partial | 33.8 | 13.1 | $74.6 \pm 0.6$ | $50.1 \pm 1.0$ | $\mathbf{80.2 \pm 2.4}$ |

Table 1: Average success rate (%) (over 4 seeds) of offline RL methods: BC, BEAR (Kumar et al., 2019), EMAQ (Ghasemipour et al., 2020), CQL (Kumar et al., 2020b) and CQL+OPAL (ours).

**Theorem 4.2.** *Let $\pi_{\psi^*}(z|s)$ be the policy obtained by CQL and let $\pi_{\psi^*,\theta}(a|s)$ refer to the policy when $\pi_{\psi^*}(z|s)$ is used together with $\pi_\theta(a|s,z)$. Let $\pi_\beta \equiv \{\pi \; ; \; \pi \sim \Pi\}$ refer to the policy generating $\mathcal{D}^r$ in MDP $\mathcal{M}$ and $z \sim \pi_\beta^H(z|s) \equiv \tau \sim \pi_{\beta,s_0=s}, z \sim q_\phi(z|\tau)$. Then, $J(\pi_{\psi^*,\theta}, M) \geq J(\pi_\beta, M) - \kappa$ with high probability $1 - \delta$ where*

$$\kappa = \mathcal{O}\left( \frac{1}{(1-\gamma^c)(1-\gamma)} \mathbb{E}_{s \sim d_{\hat{M}_H}^{\pi_{\psi^*,\theta}}(s)} \left[ \sqrt{|\mathcal{Z}|(\mathrm{D}_{\mathrm{CQL}}(\pi_{\psi^*}, \pi_\beta^H)(s) + 1)} \right] \right) \tag{9}$$

$$- \frac{\alpha}{1-\gamma^c} E_{s \sim d_{\mathcal{M}_H}^{\pi_{\psi^*}}(s)} \left[ \mathrm{D}_{\mathrm{CQL}}(\pi_{\psi^*,\theta}, \pi_\beta^H)(s) \right], \tag{10}$$

*where $\mathrm{D}_{\mathrm{CQL}}$ is a measure of the divergence between two policies; see the appendix for a formal statement.*

The precise bound along with a proof is described in Appendix B.1. Intuitively, this bound suggests that the worst-case deterioration over the learned policy depends on the divergence between the learned latent-space policy $\mathrm{D}_{\mathrm{CQL}}$ and the actual primitive distribution, which is controlled via any conservative offline RL algorithm (Kumar et al. (2020b) in our experiments) and the size of the latent space $|\mathcal{Z}|$. Crucially, note that comparing Equation 9 to the performance bound for CQL (Equation 6 in Kumar et al. (2020b)) reveals several benefits pertaining to **(1)** temporal abstraction – a reduction in the factor of horizon by virtue of $\gamma^c$, and **(2)** reduction in the amount of worst-case error propagation due to a reduced action space $|\mathcal{Z}|$ vs. $|\mathcal{A}|$. Thus, as evident from the above bound, the total error induced due to a combination of distributional shift and sampling is significantly reduced when OPAL is used as compared to the standard RL counterpart of this bound which is affected by the size of the entire action space for each and every timestep in the horizon. This formalizes our intuition that OPAL helps to partly mitigate distributional shift and sampling error. One downside of using a latent space policy is that we incur unsupervised learning error while learning primitives. However, empirically, this unsupervised learning error gets dominated by other error terms pertaining to offline RL. That is, it is much easier to control unsupervised loss than errors arising in offline RL.

## 5 EVALUATION

In this section, we will empirically show that OPAL improves learning of downstream tasks with offline RL, and then briefly show the same with few-shot imitation learning, online RL, and online multi-task transfer learning. Unless otherwise stated, we use $c = 10$ and $\dim(\mathcal{Z}) = 8$. See Appendix C for further implementation and experimental details. Visualizations and code are available at https://sites.google.com/view/opal-iclr

### 5.1 OFFLINE RL WITH OPAL

**Description:** We use environments and datasets provided in D4RL (Fu et al., 2020). Since the aim of our method is specifically to perform offline RL in settings where the offline data comprises varied and undirected multi-task behavior, we focus on Antmaze medium (diverse dataset), Antmaze large (diverse dataset), and Franka kitchen (mixed and partial datasets). The Antmaze datasets involve a simulated ant robot performing undirected navigation in a maze. The task is to use this undirected dataset to solve a specific point-to-point navigation problem, traversing the maze from one corner to the opposite corner, with only sparse 0-1 completion reward for reaching the goal. The kitchen datasets involves a franka robot manipulating multiple objects (microwave, kettle, etc.) either in an

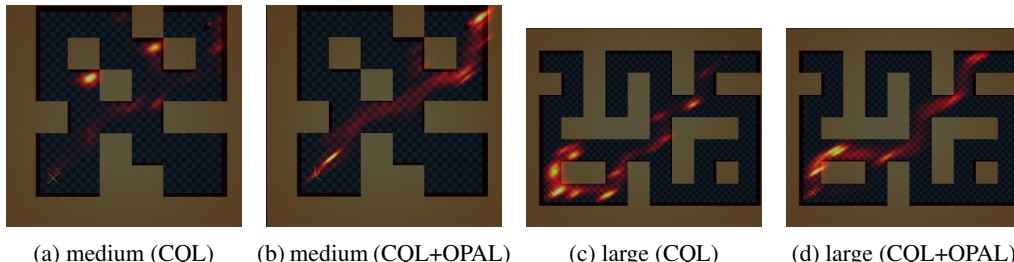

| (a) medium (CQL) | (b) medium (CQL+OPAL) | (c) large (CQL) | (d) large (CQL+OPAL) |

Figure 3: State visitation heatmaps for antmaze medium policies learned using (1) CQL and (2) CQL+OPAL, and antmaze large policies learned using (3) CQL and (4) CQL+OPAL.

| Environment | BC | BC+OPAL (ours) | BC+SVAE |
|---|---|---|---|
| antmaze medium (diverse) | $30.1 \pm 3.2$ | $\mathbf{81.5 \pm 2.7}$ | $72.8 \pm 2.3$ |
| antmaze large (diverse) | $9.2 \pm 2.5$ | $\mathbf{63.5 \pm 2.3}$ | $49.4 \pm 2.2$ |

Table 2: Average success rate (%) (over 4 seeds) of few-shot IL methods: BC, BC+OPAL, and BC+SVAE (Wang et al., 2017).

undirected manner (mixed dataset) or in a partially task directed manner (partial dataset). The task is to use the datasets to arrange objects in a desired configuration, with only sparse 0-1 completion reward for every object that attains the target configuration.

**Baseline:** We use Behavior cloning (BC), BEAR (Kumar et al., 2019), EMAQ (Ghasemipour et al., 2020), and CQL (Kumar et al., 2020b) as baselines. We compare it to CQL+OPAL, which first uses OPAL to distill primitives from the offline dataset before applying CQL to learn a primitive-directing high-level policy.

**Results:** As shown in Table 1, CQL+OPAL outperforms nearly all the baselines on antmaze (see Figure 1 and Figure 3 for visualization) and kitchen tasks, with the exception of EMAQ having similar performance on kitchen mixed. To ensure fair comparison with EMAQ, we use an autoregressive primitive policy. With the exception of EMAQ on kitchen mixed, we are not aware of any existing offline RL algorithms that achieves similarly good performance on these tasks; moreover, we are not aware of any existing *online* RL algorithms which solve these tasks (see Table 3 for some comparisons), highlighting the benefit of using offline datasets to circumvent exploration challenges. There are two potential reasons for OPAL's success. First, temporally-extended primitives could make the reward propagation learning problem easier. Second, the primitives may provide a better latent action space than the atomic actions of the environment. To understand the relative importance of these factors, we experimented with an ablation of CQL+OPAL that uses $c = 1$ to remove temporal abstraction. In this case, we find the method's performance to be similar to standard CQL. This implies that the temporal abstraction provided by OPAL is one of the main contributing factors to its good performance. This observation also agrees with our theoretical analysis. See Appendix E for detailed discussion.

## 5.2 FEW-SHOT IMITATION LEARNING WITH OPAL

**Description:** Previously, we assumed that we have access to a task reward function, but only undirected data that performs other tasks. Now, we will study the opposite case, where we are not provided with a reward function for the new task either, but instead receive a small number of task-specific demonstrations that illustrate optimal behavior. Simply imitating these few demonstrations is insufficient to obtain a good policy, and our experiments evaluate whether OPAL can effectively incorporate the prior data to enable few-shot adaptation in this setting. We use the Antmaze environments (diverse datasets) to evaluate our method and use an expert policy for these environments to sample $n = 10$ successful trajectories.

**Baseline and Results:** For baselines, we use Behavior cloning (BC) and the model from Wang et al. (2017), which prescribes using a sequential VAE (SVAE) over state trajectories in conjunction with imitation learning. As shown in Table 2, BC+OPAL clearly outperforms other baselines, show-

| Environment | HIRO | SAC+BC | SAC+OPAL(ours) | DDQN+DDCO |
|---|---|---|---|---|
| antmaze medium sparse (diverse) | 0.0 | 0.0 | **81.6 ± 3.7** | 0.0 |
| antmaze large sparse (diverse) | 0.0 | 0.0 | 0.0 | 0.0 |
| antmaze medium dense (diverse) | 0.0 | 0.0 | **81.3 ± 3.3** | 0.0 |
| antmaze large dense (diverse) | 12 | 0.0 | **81.5 ± 3.9** | 0.0 |

Table 3: Average success rate (%) (over 4 seeds) of online RL methods: HIRO (Nachum et al., 2018a), SAC+BC, SAC+OPAL, and DDQN+DDCO (Krishnan et al., 2017). These methods were ran for 2.5e6 steps for antmaze medium environments and 17.5e6 steps for antmaze large environments.

| Models | MT10 | MT50 |
|---|---|---|
| PPO | 15.2 ± 4.8 | 5.1 ± 2.2 |
| PPO+OPAL(ours) | **70.1 ± 4.3** | **45.3 ± 3.1** |
| SAC | 39.5 | 28.8 |

Table 4: Due to improved exploration, PPO+OPAL outperforms PPO and SAC on MT10 and MT50 in terms of average success rate (%) (over 4 seeds).

ing the importance of temporal abstraction and ascertaining the quality of learned primitives. See Appendix A for detailed discussion.

### 5.3 ONLINE RL AND MULTI-TASK TRANSFER WITH OPAL

**Description:** For online RL and multi-task transfer learning, we learn a task policy in space of primitives $\pi_\theta(a|s, z)$ while keeping it fixed. For multi-task transfer, the task policy also takes in the task id and we use $c = 5$ and $\mathcal{Z} = 8$. Since the primitives need to transfer to a different state distribution for multi-task transfer, it only learns the action sub-trajectory distribution and doesn't take in the state feedback. See Appendix A for a detailed description of models. For online RL, we use the Antmaze environments (diverse datasets) with sparse and dense rewards for evaluating our method. For online multi-task transfer learning, we learn primitives with expert data from pick-and-place task and then use it to learn multi-task policy for MT10 and MT50 (from metaworld (Yu et al., 2020)), containing 10 and 50 robotic manipulation tasks which needs to be solved simultaneously.

**Baseline and Results:** For online RL, we use HIRO (Nachum et al., 2018b), a state-of-the-art hierarchical RL method, SAC (Haarnoja et al., 2018) with Behavior cloning (BC) pre-training on $\mathcal{D}$, and Discovery of Continuous Options (DDCO) (Krishnan et al., 2017) which uses $\mathcal{D}$ to learn a discrete set of primitives and then learns a task policy in space of those primitives with online RL (Double DQN (DDQN) (Van Hasselt et al., 2015)). For online multi-task transfer learning, we use PPO (Schulman et al., 2017) and SAC (Haarnoja et al., 2018) as baselines. As shown in Table 3 and Table 4, OPAL uses temporal abstraction to improve exploration and thus accelerate online RL and multi-task transfer learning. See Appendix A for detailed discussion.

## 6 DISCUSSION

We proposed *Offline Primitives for Accelerating offline RL* (OPAL) as a preproccesing step for extracting recurring *primitive* behaviors from undirected and unlabelled dataset of diverse behaviors. We derived theoretical statements which describe under what conditions OPAL can improve learning of downstream offline RL tasks and showed how these improvements manifest in practice, leading to significant improvements in complex manipulation tasks. We further showed empirical demonstrations of OPAL's application to few-shot imitation learning, online RL, and online multi-task transfer learning. In this work, we focused on simple auto-encoding models for representing OPAL, and an interesting avenue for future work is scaling up this basic paradigm to image-based tasks.

## 7 ACKNOWLEDGEMENTS

We would like to thank Ben Eysenbach and Kamyar Ghasemipour for valuable discussions at different points over the course of this work. This work was supported by Google, DARPA Machine Common Sense grant and MIT-IBM grant.

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

# Appendices

## A  OTHER APPLICATIONS OF OPAL

### A.1  FEW-SHOT IMITATION LEARNING WITH OPAL

**Additional Assumption**: In addition to $\mathcal{D}$, we assume access to a small number of expert demonstrations $\mathcal{D}^{\mathrm{exp}} = \{\tau_i := (s_t, a_t)_{t=0}^{T-1}\}_{i=1}^n$ where $n \ll N$.

**How to use with OPAL?** In imitation learning, the aim is to recover an expert policy given a small number of stochastically sampled expert demonstrations $\mathcal{D}^{\mathrm{exp}} = \{\tau_i := (s_t, a_t)_{t=0}^{T-1}\}_{i=1}^n$. As in the offline RL setting, we use the primitives $\pi_\theta(a|s, z)$ learned by OPAL as a low-level controller and learn a high-level policy $\pi_\psi(z|s)$. We first partition the expert demonstrations into sub-trajectories $\mathcal{D}_{\mathrm{par}}^{\mathrm{exp}} = \{\tau_{i,k} := (s_{k+t}, a_{k+t})_{t=0}^{c-1} \text{ for } k = 0, \dots, T-c\}_{i=1}^n$ of length $c$. We then use the learned encoder $q_\phi(z|\tau)$ to label these sub-trajectories with latent actions $z_{i,k} \sim q_\phi(z|\tau_{i,k})$ and thus create a dataset $\mathcal{D}_{\mathrm{hi}}^{\mathrm{exp}} = \{(s_{k+t}^i, z_{i,k}) \text{ for } k = 0, \dots, T-c\}_{i=1}^n$. We use $\mathcal{D}_{\mathrm{hi}}^{\mathrm{exp}}$ to learn the high-level policy $\pi_\psi(z|s)$ using behavioral cloning. As in the offline RL setting, we also finetune $\pi_\theta(a|s, z)$ with latent-conditioned behavioral cloning to ensure consistency of the labelled latents.

**Evaluation Description:** We receive a small number of task-specific demonstrations that illustrate optimal behavior. Simply imitating these few demonstrations is insufficient to obtain a good policy, and our experiments evaluate whether OPAL can effectively incorporate the prior data to enable few-shot adaptation in this setting. We use the Antmaze environments (diverse datasets) to evaluate our method and use an expert policy for these environments to sample $n = 10$ successful trajectories.

**Baseline:** We evaluate two baselines. First, we test a simple behavioral cloning (BC) baseline, which trains using a max-likelihood loss on the expert data. In order to make the comparison fair to OPAL (which uses the offline dataset $\mathcal{D}$ in addition to the expert dataset), we pretrain the BC agent on the undirected dataset using the same max-likelihood loss. As a second baseline and to test the quality of OPAL-extracted primitives, we experiment with an alternative unsupervised objective from Wang et al. (2017), which prescribes using a sequential VAE (SVAE) over state trajectories in conjunction with imitation learning.

**Results:** As shown in Table 2, BC+OPAL clearly outperforms other baselines, showing the importance of temporal abstraction and ascertaining the quality of learned primitives. SVAE's slightly worse performance suggests that decoding the state trajectory directly is more difficult than simply predicting the actions, as OPAL does, and that this degrades downstream task learning.

### A.2  ONLINE RL WITH OPAL

**Additional Assumptions**: We assume online access to $\mathcal{M}$; i.e., access to Monte Carlo samples of episodes from $\mathcal{M}$ given an arbitrary policy $\pi$.

**How to use with OPAL?** To apply OPAL to standard online RL, we fix the learned primitives $\pi_\theta(a|s, z)$ and learn a high-level policy $\pi_\psi(z|s)$ in an online fashion using the latents $z$ as temporally-extended actions. Specifically, when interacting the environment, $\pi_\psi(z|s)$ chooses an appropriate primitive every $c$ steps, and this primitive acts on the environment directly for $c$ timesteps. Any off-the-shelf online RL algorithm can be used to learn $\psi$. In our experiments, we use SAC (Haarnoja et al., 2018). To ensure that $\pi_\psi(z|s)$ stays close to the data distribution and avoid generalization issues associated with the fixed $\pi_\theta(a|s, z)$, we add an additional KL penalty to the reward of the form $D_{\mathrm{KL}}(\pi_\psi(z|s)||\rho_\omega(z|s_0))$.

**Evaluation Description:** We use Antmaze medium (diverse dataset) and Antmaze large (diverse dataset) from the D4RL task suite (Fu et al., 2020) to evaluate our method. We evaluate using both a dense distance-based reward $-\|g - \mathrm{ant}_{\mathrm{xy}}\|$ and a sparse success-based reward $\mathbb{1}[\|g - \mathrm{ant}_{\mathrm{xy}}\| \leq 0.5]$ (the typical default for this task), where $\mathrm{ant}_{\mathrm{xy}}$ is the 2d position of the ant in the maze.

**Baseline:** To solve these tasks through online RL, we need both (i) hierarchy (i.e. learning a policy on top of primitives) which improves exploration (Nachum et al., 2019b) and (ii) unlabelled (i.e. no task reward) offline dataset which allows us to bootstrap the primitives. This informed our choice

of these three baselines. First, to test the role of $\mathcal{D}$ in exploration, we use HIRO (Nachum et al., 2018b), a state-of-the-art hierarchical RL method, as a baseline. Second, to test the role of temporal abstraction, we pre-train a flat policy on $\mathcal{D}$ using behavioral cloning (BC) and then finetune the policy on downstream tasks with SAC. Third, to test the quality of extracted primitives for online RL, we extract a discrete set of primitives with Deep Discovery of Continuous Options (DDCO) (Krishnan et al., 2017) and use Double DQN (DDQN) (Van Hasselt et al., 2015) to learn a task policy in the space of learned discrete primitives.

**Results:** As shown in Table 3, SAC+OPAL outperforms all the baselines, showing (1) the importance of $\mathcal{D}$ in exploration, (2) the role of temporal abstraction, and (3) the good quality of learned primitives. Except for HIRO on Antmaze large with dense rewards, all other baselines fail to make any progress at all. In contrast, SAC+OPAL only fails to make progress on Antmaze large with sparse rewards.

### A.3 ONLINE MULTI-TASK TRANSFER LEARNING WITH OPAL

**Additional Assumption**: We assume the existence of $M$ additional MDPs $\{\mathcal{M}_i = (\mathcal{S}_i, \mathcal{A}, \mathcal{P}_i, r_i, \gamma)\}_{i=1}^M$ where the action space $\mathcal{A}$, and the discount factor $\gamma$ are same as those of $\mathcal{M}$.

**How to use it with OPAL?** In the multi-task setting, we aim to learn near-optimal behavior policies on $M$ MDPs $\{\mathcal{M}_i = (\mathcal{S}_i, \mathcal{A}, \mathcal{P}_i, r_i, \gamma)\}_{i=1}^M$. As in the previous applications of OPAL, we learn a set of high-level policies $\pi_\psi(z|s, i)$ which will direct pretrained primitives $\pi_\theta(a|s, z)$ to maximize cumulative rewards. Since the state space in the $M$ MDPs is potentially distinct from that in the offline dataset $\mathcal{D}$, we cannot transfer the state distribution and can only hope to transfer the action sub-trajectory distribution. Therefore, during the unsupervised training phase for learning $\pi_\theta$, we make the encoder and the decoder blind to the states in sub-trajectory. Specifically, the encoder becomes $(\mu_z^{enc}, \sigma_z^{enc}) = q_\phi(z_t|s_t, \{a_{t+i}\}_{i=0}^{c-1})$ and is represented by a bidirectional GRU. The decoder becomes $\pi_\theta(\{a_t, \ldots, a_{t+c-1}\}|z_t)$ which decodes the entire action sub-trajectory from the latent vector and is represented by a GRU. With these state-agnostic primitives in hand, we then learn a policy $\pi_\psi(z|s, i)$ using any off-the-shelf online RL method. In our experiments, we use Proximal Policy Optimization (PPO) (Schulman et al., 2017).

**Evaluation Description:** We use the Metaworld task suite (Yu et al., 2020) to evaluate our method. The dataset $\mathcal{D}$ for learning primitives consists of trajectories generated by an expert policy for a goal-conditioned pick-and-place task. The pick-and-place task is suitable for unsupervised primitive learning because it contains all the basic operations (eg: move, grasp, place) required for performing more complex manipulation tasks in the Metaworld. Once we have learned the primitives, we learn a policy $\pi_\psi(z|s, i)$ for the MT10 and MT50 benchmarks, where MT10 and MT50 contain 10 and 50 robotic manipulation tasks, respectively, which we need to be solved simultaneously. In these experiments we use $c = 5$ and $\dim(\mathcal{Z}) = 8$.

**Baseline:** We use SAC (Haarnoja et al., 2018) and PPO (Schulman et al., 2017) as baselines.

**Results:** As shown in Table 4, PPO+OPAL clearly outperforms both PPO and SAC, showing the importance of temporal abstraction in online multi-task transfer.

## B PROOF OF THEOREMS

### B.1 BOUNDING THE SUBOPTIMALITY OF THE LEARNED PRIMITIVES

We will begin by proving the following lemma which bounds the sampling error incurred by $J(\phi, \theta, \omega)$.

**Lemma B.0.1.** *With high probability $1 - \delta$,*

$$\left| J(\theta, \phi, \omega) - \mathbb{E}_{\pi \sim \Pi, \tau \sim \pi, z \sim q_\phi(z|\tau)} \left[ -\sum_{t=0}^{c-1} \log \pi_\theta(a_t|s_t, z) \right] \right| \leq \sqrt{\frac{S_J}{\delta}} \tag{11}$$

*where $S_J$ is a constant dependent on concentration properties of $\pi_\theta(a|s, z)$ and $q_\phi(z|\tau)$.*

*Proof.* To be concise, let us denote the sampling error in $J(\phi, \theta, \omega)$ by

$$\Delta_J = \left| J(\theta, \phi, \omega) - \mathbb{E}_{\pi \sim \Pi, \tau \sim \pi, z \sim q_\phi(z|\tau)} \left[ -\sum_{t=0}^{c-1} \log \pi_\theta(a_t|s_t, z) \right] \right| \tag{12}$$

Applying Chebyshev's inequality to $\Delta_J$, we get that, with high probability $1 - \delta$,

$$\Delta_J \leq \sqrt{\frac{\mathrm{Var}_{\pi \sim \Pi, \tau \sim \pi, z \sim q_\phi(z|\tau)}(-\sum_{t=0}^{c-1} \log \pi_\theta(a_t|s_t, z))}{\delta}} = \sqrt{\frac{S_J}{\delta}} \tag{13}$$

$\square$

Therefore, combining all the equations, we have

$$\mathbb{E}_{\pi \sim \Pi, \tau \sim \pi, z \sim q_\phi(z|\tau)} \left[ -\sum_{t=0}^{c-1} \log \pi_\theta(a_t|s_t, z) \right] \leq J(\theta, \phi, \omega) + \sqrt{\frac{S_J}{\delta}} \tag{14}$$

We present a general performance difference lemma that will help in our proof of Lemma 4.0.1.

**Lemma B.0.2.** *If $\pi_1$ and $\pi_2$ are two policies in $\mathcal{M}$, then*

$$|J_{\mathrm{RL}}(\pi_1, \mathcal{M}) - J_{\mathrm{RL}}(\pi_2, \mathcal{M})| \leq \frac{2}{(1-\gamma)^2} \mathrm{R}_{\max} \mathbb{E}_{s \sim d^{\pi_1}}[\mathrm{D}_{\mathrm{TV}}(\pi_1(a|s)\|\pi_2(a|s))]. \tag{15}$$

*Proof.* Following the derivations in Achiam et al. (2017) and Nachum et al. (2018a), we express the performance of a policy $\pi$ in $\mathcal{M}$ in terms of linear opterators:

$$J_{\mathrm{RL}}(\pi, \mathcal{M}) = (1-\gamma)^{-1} R^\top (I - \gamma \Pi_\pi \mathcal{P})^{-1} \Pi_\pi \mu, \tag{16}$$

where $R$ is a vector representation of the rewards of $\mathcal{M}$, $\Pi_\pi$ is a linear operator mapping state distributions to state-action distributions according to $\pi$, and $\mu$ is a vector representation of the initial state distribution of $\mathcal{M}$. Accordingly, we express the performance difference of $\pi_1, \pi_2$ as,

$$|J_{\mathrm{RL}}(\pi_1, \mathcal{M}) - J_{\mathrm{RL}}(\pi_2, \mathcal{M})| = |R^\top((I - \gamma \Pi_1 \mathcal{P})^{-1} \Pi_1 \mu - (I - \gamma \Pi_2 \mathcal{P})^{-1} \Pi_2 \mu)| \tag{17}$$

$$\leq \mathrm{R}_{\max} |(I - \gamma \Pi_1 \mathcal{P})^{-1} \Pi_1 \mu - (I - \gamma \Pi_2 \mathcal{P})^{-1} \Pi_2 \mu|. \tag{18}$$

By the triangle inequality, we may bound equation 18 by

$$\mathrm{R}_{\max} \big( |(I - \gamma \Pi_1 \mathcal{P})^{-1} \Pi_1 \mu - (I - \gamma \Pi_2 \mathcal{P})^{-1} \Pi_1 \mu| +$$
$$|(I - \gamma \Pi_2 \mathcal{P})^{-1} \Pi_1 \mu - (I - \gamma \Pi_2 \mathcal{P})^{-1} \Pi_2 \mu| \big). \tag{19}$$

We begin by approaching the first term inside the parentheses of equation 19. That first term may be expressed as

$$|(I - \gamma \Pi_2 \mathcal{P})^{-1}(I - \gamma \Pi_2 \mathcal{P} - (I - \gamma \Pi_1 \mathcal{P}))(I - \gamma \Pi_1 \mathcal{P})^{-1} \Pi_1 \mu| \tag{20}$$

$$= |\gamma(I - \gamma \Pi_2 \mathcal{P})^{-1}(\Pi_1 - \Pi_2)\mathcal{P}(I - \gamma \Pi_1 \mathcal{P})^{-1} \Pi_1 \mu| \tag{21}$$

$$\leq \frac{\gamma}{1-\gamma} |(\Pi_1 - \Pi_2)\mathcal{P}(I - \gamma \Pi_1 \mathcal{P})^{-1} \Pi_1 \mu| \tag{22}$$

$$= \frac{2\gamma}{(1-\gamma)^2} \mathbb{E}_{s \sim (1-\gamma)\mathcal{P}(I-\gamma \Pi_1 \mathcal{P})^{-1} \Pi_1 \mu}[\mathrm{D}_{\mathrm{TV}}(\pi_1(a|s)\|\pi_2(a|s))]. \tag{23}$$

Now we continue to the second term inside the parentheses of equation 19. This second term may be expressed as

$$|(I - \gamma \Pi_2 \mathcal{P})^{-1}(\Pi_1 - \Pi_2)\mu| \leq \frac{1}{1-\gamma} |(\Pi_1 - \Pi_2)\mu| \tag{24}$$

$$= \frac{2}{1-\gamma} \mathbb{E}_{s \sim \mu}[\mathrm{D}_{\mathrm{TV}}(\pi_1(a|s)\|\pi_2(a|s))]. \tag{25}$$

To combine equations 23 and 25, we note that

$$d^{\pi_1} = \gamma \cdot (1 - \gamma)\mathcal{P}(I - \gamma\Pi_1\mathcal{P})^{-1}\Pi_1\mu + (1 - \gamma) \cdot \mu. \tag{26}$$

Thus, we have

$$|J_{\text{RL}}(\pi_1, \mathcal{M}) - J_{\text{RL}}(\pi_2, \mathcal{M})| \leq \frac{2}{(1 - \gamma)^2}R_{\max}\mathbb{E}_{s \sim d^{\pi_1}}[D_{\text{TV}}(\pi_1(a|s)||\pi_2(a|s))], \tag{27}$$

as desired. $\qquad\qquad\square$

**Lemma 4.0.1.** *If $\pi_1$ and $\pi_2$ are two policies in $\mathcal{M}$, then*

$$|J_{\text{RL}}(\pi_1, \mathcal{M}) - J_{\text{RL}}(\pi_2, \mathcal{M})| \leq \frac{2}{(1 - \gamma^c)(1 - \gamma)}R_{\max}\mathbb{E}_{s \sim d_c^{\pi_1}}[D_{\text{TV}}(\pi_1(\tau|s)||\pi_2(\tau|s))], \tag{28}$$

*where $D_{\text{TV}}(\pi_1(\tau|s)||\pi_2(\tau|s))$ denotes the TV divergence over $c$-length sub-trajectories $\tau$ sampled from $\pi_1$ vs. $\pi_2$ (see section 3). Furthermore,*

$$\text{SubOpt}(\theta) \leq \frac{2}{(1 - \gamma^c)(1 - \gamma)}R_{\max}\mathbb{E}_{s \sim d_c^{\pi^*}}[D_{\text{TV}}(\pi^*(\tau|s)||\pi_{\theta,\psi^*}(\tau|s))]. \tag{29}$$

*Proof.* We focus on proving equation 28, as the subsequent derivation of equation 29 is straightforward by definition of SubOpt.

To derive equation 28, we may simply consider $\pi_1$ and $\pi_2$ acting in an "every-$c$-steps" version of $\mathcal{M}$, where the action space is now $\tau$ and reward are accumulated over $c$ steps using $\gamma$-discounting. Note that in this abstracted version of $\mathcal{M}$, the max reward is $\frac{1-\gamma^c}{1-\gamma}R_{\max}$ and the MDP discount is $\gamma^c$. Plugging this into the result of Lemma B.0.2 immediately yields the desired claim. $\qquad\square$

**Theorem 4.1.** *Let $\theta, \phi, \omega$ be the outputs of solving equation 1, such that $J(\theta, \phi, \omega) = \epsilon_c$. Then, with high probability $1 - \delta$, for any $\overline{\pi}$ that is $\zeta$-common in $\Pi$, there exists a distribution $H$ over $z$ such that for $\pi_\theta^H(\tau|s) := \mathbb{E}_{z \sim H}[\pi_\theta(\tau|z, s)]$,*

$$\mathbb{E}_{s \sim \kappa}[D_{\text{TV}}(\overline{\pi}(\tau|s)||\pi_\theta^H(\tau|s))] \leq \zeta + \sqrt{\frac{1}{2}\left(\epsilon_c + \sqrt{\frac{S_J}{\delta}} + \mathcal{H}_c\right)}, \tag{30}$$

*where $\mathcal{H}_c = \mathbb{E}_{\pi \sim \Pi, s \sim \kappa, \tau \sim \pi}[\sum_{t=0}^{c-1}\log\pi(a_t|s_t)]$ (i.e. a constant and property of $\mathcal{D}$) and $S_J$ is a positive constant incurred due to sampling error in $J(\theta, \phi, \omega)$ and depends on concentration properties of $\pi_\theta(a|s, z)$ and $q_\phi(z|\tau)$.*

*Proof.* We start with application of the triangle inequality:

$$D_{\text{TV}}(\overline{\pi}(\tau|s)||\pi_\theta^H(\tau|s)) \leq D_{\text{TV}}(\overline{\pi}(\tau|s)||\pi(\tau|s)) + D_{\text{TV}}(\pi(\tau|s)||\pi_\theta^H(\tau|s)). \tag{31}$$

Taking expectation with respect to $\pi \sim \Pi, s \sim \kappa$ on both the sides, we get

$$\mathbb{E}_{s \sim \kappa}[D_{\text{TV}}(\overline{\pi}(\tau)||\pi_\theta^H(\tau))] \leq \mathbb{E}_{\pi \sim \Pi, s \sim \kappa}[D_{\text{TV}}(\overline{\pi}(\tau)||\pi(\tau))] + \mathbb{E}_{\pi \sim \Pi, s \sim \mu}[D_{\text{TV}}(\pi(\tau)||\pi_\theta^H(\tau))] \tag{32}$$

$$\leq \mathbb{E}_{\pi \sim \Pi, s \sim \kappa}[D_{\text{TV}}(\overline{\pi}(\tau)||\pi(\tau))] \tag{33}$$

$$+ \mathbb{E}_{\pi \sim \Pi, s \sim \mu}[\sqrt{\frac{1}{2}D_{\text{KL}}(\pi(\tau)||\pi_\theta^H(\tau))}] \tag{34}$$

$$\leq \zeta + \sqrt{\frac{1}{2}\mathbb{E}_{\pi \sim \Pi, \tau \sim \pi, s \sim \kappa}[\log\pi(\tau) - \log\mathbb{E}_{z \sim H}[\pi_\theta(\tau|z)]]} \tag{35}$$

$$\leq \zeta + \sqrt{\frac{1}{2}\mathbb{E}_{\pi \sim \Pi, s \sim \kappa, \tau \sim \pi, z \sim H}[\log\pi(\tau) - \log\pi_\theta(\tau|z)]}. \tag{36}$$

The last two inequality used Jensen's inequality. Let $H(z) = \mathbb{E}_{\pi \sim \Pi, \tau \sim \pi}[q_\phi(z|\tau)]$. Cancelling out the dynamics and using equation 14 we get,

$$\mathbb{E}_{s \sim \kappa}[\mathrm{D}_{\mathrm{TV}}(\overline{\pi}(\tau)||\pi_\theta^H(\tau))]$$

$$\leq \zeta + \sqrt{\frac{1}{2}\mathbb{E}_{\pi \sim \Pi, s \sim \kappa, \tau \sim \pi, z \sim q_\phi(z|\tau)}\left[\sum_{t=0}^{c-1}(\log \pi(a_t|s_t) - \log \pi_\theta(a_t|s_t, z))\right]}$$

$$\leq \zeta + \sqrt{\frac{1}{2}\left(\mathcal{H}_c + \sqrt{\frac{S_J}{\delta}} + J(\theta, \phi, \omega)\right)}$$

$$= \zeta + \sqrt{\frac{1}{2}\left(\mathcal{H}_c + \epsilon_c + \sqrt{\frac{S_J}{\delta}}\right)}$$

$\square$

**Corollary 4.1.1.** *If the optimal policy $\pi^*$ of $\mathcal{M}$ is $\zeta$-common in $\Pi$, and $\left\|\frac{d_c^{\pi^*}}{\kappa}\right\|_\infty \leq \xi$, then, with high probability $1 - \delta$,*

$$\mathrm{SubOpt}(\theta) \leq \frac{2\xi}{(1-\gamma^c)(1-\gamma)}\mathrm{R}_{\max}\left(\zeta + \sqrt{\frac{1}{2}\left(\epsilon_c + \sqrt{\frac{S_J}{\delta}} + \mathcal{H}_c\right)}\right) \tag{37}$$

*Proof.* Expanding lemma 4.0.1 using the above assumption, we have

$$\mathrm{SubOpt}(\theta) \leq |J_{\mathrm{RL}}(\pi^*, \mathcal{M}) - J_{\mathrm{RL}}(\pi_\theta^H, \mathcal{M})| \tag{38}$$

$$\leq \frac{2}{(1-\gamma^c)(1-\gamma)}\mathrm{R}_{\max}\mathbb{E}_{s \sim d_c^{\pi^*}}[\mathrm{D}_{\mathrm{TV}}(\pi^*(\tau|s)||\pi_\theta^H(\tau|s))] \tag{39}$$

$$\leq \frac{2}{(1-\gamma^c)(1-\gamma)}\mathrm{R}_{\max}\left\|\frac{d_c^{\pi^*}}{\kappa}\right\|_\infty \mathbb{E}_{s \sim \kappa}[\mathrm{D}_{\mathrm{TV}}(\pi^*(\tau|s)||\pi_\theta^H(\tau|s))] \tag{40}$$

$$\leq \frac{2\xi}{(1-\gamma^c)(1-\gamma)}\mathrm{R}_{\max}\mathbb{E}_{s \sim \kappa}[\mathrm{D}_{\mathrm{TV}}(\pi^*(\tau|s)||\pi_\theta^H(\tau|s))] \tag{41}$$

Now, we can use theorem 4.1 to prove the corollary. $\square$

### B.2 PERFORMANCE BOUNDS FOR OPAL

**Theorem 4.2.** *Let $\pi_{\psi^*}(z|s)$ be the policy obtained by CQL and let $\pi_{\psi^*, \theta}(a|s)$ refer to the policy when $\pi_{\psi^*}(z|s)$ is used together with $\pi_\theta(a|s, z)$. Let $\pi_\beta \equiv \{\pi ; \pi \sim \Pi\}$ refer to the policy generating $\mathcal{D}^r$ in MDP $\mathcal{M}$ and $z \sim \pi_\beta^H(z|s) \equiv \tau \sim \pi_{\beta, s_0 = s}, z \sim q_\phi(z|\tau)$. Then, $J(\pi_{\psi^*, \theta}, M) \geq J(\pi_\beta, M) - \kappa$ with high probability $1 - \delta$ where*

$$\kappa = \mathcal{O}\left(\frac{1}{(1-\gamma^c)(1-\gamma)}\mathbb{E}_{s \sim d_{\hat{M}_H}^{\pi_{\psi^*, \theta}}(s)}\left[\sqrt{|\mathcal{Z}|(\mathrm{D}_{\mathrm{CQL}}(\pi_{\psi^*}, \pi_\beta^H)(s) + 1)}\right]\right) \tag{42}$$

$$- \frac{\alpha}{1-\gamma^c}E_{s \sim d_{\mathcal{M}_H}^{\pi_{\psi^*}}(s)}\left[\mathrm{D}_{\mathrm{CQL}}(\pi_{\psi^*, \theta}, \pi_\beta^H)(s)\right] \tag{43}$$

*Proof.* We assume that the variational posterior $q_\phi(z|\tau)$ obtained after learning OPAL from $\mathcal{D}$ is same (or nearly same) as the true posterior $p(z|\tau)$. $q_\phi$ can be used to define $\pi_\beta^H(z|s)$ as $z \sim \pi_\beta^H \equiv \tau \sim \pi_\beta, z \sim q_\phi(z|\tau)$. This induces an MDP $\mathcal{M}_H = (\mathcal{S}, \mathcal{Z}, \mathcal{P}_z, r_z, \gamma^c)$ where $\mathcal{Z}$ is the inferred latent space for choosing primitives, $\mathcal{P}_z$ and $r_z$ are the latent dynamics and reward function such that $s_{t+c} \sim \mathcal{P}_z(s_{t+c}|s_t, z_t) \equiv s_{t+i+1} \sim \mathcal{P}(s_{t+i+1}|s_{t+i}, a_{t+i}), a_{t+i} \sim \pi(a_{t+i}|s_{t+i}, z_t) \forall i \in \{0, 1, \ldots, c-1\}$ and $r_z(s_t, z_t) = \sum_{i=0}^{c-1}\gamma^i r(s_{t+i}, a_{t+i})$, and $\gamma^c$ is the new discount factor effectively reducing the task horizon by a factor of $c$. $\pi(a|s, z)$ is the primitive induced by $q_\phi$ and $\pi_\beta$. Since $q_\phi$ captures the true posterior, $\pi(a|s, z)$ is the optimal primitive you can learn and its

autoencoding loss, under true expectation, is $\epsilon_c^* = \mathbb{E}_{\pi \sim \Pi, \tau \sim \pi, z \sim q_\phi(z|\tau)} \left[ -\sum_{t=0}^{c-1} \log \pi(a_t|s_t, z) \right]$. Therefore, $\tau \sim \pi_\beta \equiv z \sim \pi_\beta^H, \tau \sim \pi(\cdot|\cdot, z)$. $\pi_\beta$ is used to collect the data $\mathcal{D}^r$ which induces an empirical MDP. We refer to the empirical MDP induced by $\mathcal{D}^r$ as $\hat{\mathcal{M}} = (\mathcal{S}, \mathcal{A}, \hat{\mathcal{P}}, \hat{\mu}, r, \gamma)$ where $\hat{\mathcal{P}}(\hat{s}'|\hat{s}, \hat{a}) = \frac{\sum_{(s,a,s') \sim \mathcal{D}} 1[s=\hat{s}, a=\hat{a}, s'=\hat{s}']}{\sum_{(s,a) \sim \mathcal{D}} 1[s=\hat{s}, a=\hat{a}]}$ and $\hat{\mu}(\hat{s}) = \frac{\sum_{s_0 \sim \mathcal{D}} 1[s_0=\hat{s}]}{N}$. We use $q_\phi$ to get $\mathcal{D}_{\text{hi}}^r$ from $\mathcal{D}^r$ which induces another empirical MDP $\hat{\mathcal{M}}_H$. Using these definitions, we will try to bound $|J(\pi_{\psi^*, \theta}, \mathcal{M}) - J(\pi_\beta, \mathcal{M})|$.

Let's break $|J(\pi_{\psi^*, \theta}, \mathcal{M}) - J(\pi_\beta, \mathcal{M})|$ into

$$|J(\pi_{\psi^*, \theta}, \mathcal{M}) - J(\pi_\beta, \mathcal{M})| \leq |J(\pi_{\psi^*, \theta}, \mathcal{M}) - J(\pi_{\psi^*}, \mathcal{M}_H)| \tag{44}$$

$$+ |J(\pi_{\psi^*}, \mathcal{M}_H - J(\pi_\beta^H, \mathcal{M}_H)| \tag{45}$$

$$+ |J(\pi_\beta^H, \mathcal{M}_H) - J(\pi_\beta, \mathcal{M})| \tag{46}$$

Since $q_\phi$ captures the true variational posterior, $\tau \sim \pi_\beta \equiv z \sim \pi_\beta^H, \tau \sim \pi(\cdot|\cdot, z)$ and therefore, $|J(\pi_\beta^H, \mathcal{M}_H) - J(\pi_\beta, \mathcal{M})| = 0$. For bounding, $|J(\pi_{\psi^*}, \mathcal{M}_H - J(\pi_\beta^H, \mathcal{M}_H)|$, we use theorem 3.6 from Kumar et al. (2020b) and apply it to $\mathcal{M}_H$ to get

$$|J(\pi_{\psi^*}, \mathcal{M}_H) - J(\pi_\beta^H, \mathcal{M}_H)| \tag{47}$$

$$\leq 2 \left( \frac{C_{r,\delta}}{1 - \gamma^c} + \frac{\gamma^c R_{\max} C_{\mathcal{P}, \delta}}{(1 - \gamma^c)(1 - \gamma)} \right) \mathbb{E}_{s \sim d_{\hat{\mathcal{M}}_H}^{\pi_{\psi^*, \theta}}(s)} \left[ \sqrt{\frac{|\mathcal{Z}|}{|\mathcal{D}(s)|}} D_{\text{CQL}}(\pi_{\psi^*}, \pi_\beta^H)(s) + 1 \right] \tag{48}$$

$$- \frac{\alpha}{1 - \gamma^c} E_{s \sim d_{\mathcal{M}_H}^{\pi_{\psi^*}}(s)} \left[ D_{\text{CQL}}(\pi_{\psi^*, \theta}, \pi_\beta^H)(s) \right] = \kappa_2 \tag{49}$$

Now, we will try to bound $|J(\pi_{\psi^*, \theta}, \mathcal{M}) - J(\pi_{\psi^*}, \mathcal{M}_H)|$. The only difference between the two is that the primitive $\pi_\theta(a|s, z)$ is used in $\mathcal{M}$ and the primitive $\pi(a|s, z)$ is used in $\mathcal{M}_H$. Therefore, we can write the above bound as $|J(\pi_{\psi^*, \theta}, \mathcal{M}) - J(\pi_{\psi^*, \pi(\cdot|\cdot, z)}, \mathcal{M})|$. Let's first bound their value function at a particular state $s$. Using Lemma 4.0.1, we get

$$|J(\pi_{\psi^*, \theta}, \mathcal{M}) - J(\pi_{\psi^*, \pi(\cdot|\cdot, z)}, \mathcal{M})| \tag{50}$$

$$\leq \frac{2R_{\max}}{(1 - \gamma^c)(1 - \gamma)} \mathbb{E}_{s \sim d_c^{\pi_{\psi^*, \pi(\cdot|\cdot, z)}}} \left[ D_{\text{TV}}(\pi_{\psi^*, \pi(\cdot|\cdot, z)}(\tau|s) || \pi_{\psi^*, \theta}(\tau|s)) \right] \tag{51}$$

$$\leq \frac{2R_{\max}}{(1 - \gamma^c)(1 - \gamma)} E_{s \sim d_c^{\pi_{\psi^*, \pi(\cdot|\cdot, z)}}} \left[ \sqrt{\frac{1}{2} D_{\text{KL}}(\pi_{\psi^*, \pi(\cdot|\cdot, z)}(\tau|s) || \pi_{\psi^*, \theta}(\tau|s))} \right] \tag{52}$$

Now, we will try to bound $D_{\text{KL}}(\pi_{\psi^*, \pi(\cdot|\cdot, z)}(\tau|s) || \pi_{\psi^*, \theta}(\tau|s))$. We have

$$\mathbb{E}_{z \sim \pi_{\psi^*}(z|s), \tau \sim \pi(\tau|s, z)} \left[ \log \frac{\pi_{\psi^*}(z|s) \prod_{t=1}^{c-1} \mathcal{P}(s_t|s_{t-1}, a_{t-1}) \prod_{t=0}^{c-1} \pi(a_t|s_t, z)}{\pi_{\psi^*}(z|s) \prod_{t=1}^{c-1} \mathcal{P}(s_t|s_{t-1}, a_{t-1}) \prod_{t=0}^{c-1} \pi_\theta(a_t|s_t, z)} \right] \tag{53}$$

$$= \mathbb{E}_{z \sim \pi_{\psi^*}(z|s), \tau \sim \pi(\tau|s, z)} \left[ \sum_{t=0}^{c-1} \log \pi(a_t|s_t, z) - \log \pi_\theta(a_t|s_t, z) \right] \tag{54}$$

$$= \mathbb{E}_{z \sim \pi_\beta(z|s), \tau \sim \pi(\tau|s, z)} \left[ \left( \frac{\pi_{\psi^*}(z|s)}{\pi_\beta^H(z|s)} \right) \sum_{t=0}^{c-1} \log \pi(a_t|s_t, z) - \log \pi_\theta(a_t|s_t, z) \right] \tag{55}$$

$$\leq \left| \frac{\pi_{\psi^*}(z|s)}{\pi_\beta^H(z|s)} \right|_\infty \mathbb{E}_{z \sim \pi_\beta(z|s), \tau \sim \pi(\tau|s, z)} \left[ \sum_{t=0}^{c-1} \log \pi(a_t|s_t, z) - \log \pi_\theta(a_t|s_t, z) \right] \tag{56}$$

$$\leq \left| \frac{\pi_{\psi^*}(z|s)}{\pi_\beta^H(z|s)} \right|_\infty \left( \epsilon_c - \epsilon_c^* + \sqrt{\frac{S_J}{\delta}} \right) \tag{57}$$

The last equation comes from above definition of $\epsilon_c^*$ and equation 14. We will now try to bound $\left|\frac{\pi_{\psi^*}(z|s)}{\pi_\beta^H(z|s)}\right|_\infty$ using $D_{\mathrm{CQL}}(\pi_{\psi^*}, \pi_\beta^H)(s)$. Using definition of $D_{\mathrm{CQL}}(\pi_{\psi^*}, \pi_\beta^H)(s)$, we have

$$D_{\mathrm{CQL}}(\pi_{\psi^*}, \pi_\beta^H)(s) = \sum_z \pi_{\psi^*}(z|s) \left( \frac{\pi_{\psi^*}(z|s)}{\pi_\beta^H(z|s)} - 1 \right) \tag{58}$$

$$\Rightarrow D_{\mathrm{CQL}}(\pi_{\psi^*}, \pi_\beta^H)(s) + 1 = \sum_z \pi_\beta^H(z|s) \left( \frac{\pi_{\psi^*}(z|s)}{\pi_\beta^H(z|s)} \right)^2 \leq \left| \frac{\pi_{\psi^*}(z|s)}{\pi_\beta^H(z|s)} \right|_\infty^2 \pi_\beta^H(\bar{z}|s) \tag{59}$$

where $\bar{z} = \arg\max_z \left( \frac{\pi_{\psi^*}(z|s)}{\pi_\beta^H(z|s)} \right)$. To be concise, let

$$\Delta_c = \left( \epsilon_c - \epsilon_c^* + \sqrt{\frac{S_J}{\delta}} \right) \tag{60}$$

Combining above equations, we have

$$D_{\mathrm{KL}}(\pi_{\psi^*}(z|s)\pi(\tau|s,z) || \pi_{\psi^*}(z|s)\pi_\theta(\tau|s,z)) \leq \Delta_c \sqrt{\frac{D_{\mathrm{CQL}}(\pi_{\psi^*}, \pi_\beta^H)(s) + 1}{\pi_\beta^H(\bar{z}|s)}} \tag{61}$$

Using this to bound the returns, we get

$$\left| J(\pi_{\psi^*,\theta}, \mathcal{M}) - J(\pi_{\psi^*, \pi(\cdot|\cdot,z)}, \mathcal{M}) \right| \tag{62}$$

$$\leq \frac{2\mathrm{R}_{\max}}{(1-\gamma^c)(1-\gamma)} E_{s \sim d_c^{\pi_{\psi^*}, \pi(\cdot|\cdot,z)}} \left[ \left( \frac{D_{\mathrm{CQL}}(\pi_{\psi^*}, \pi_\beta^H)(s) + 1}{\pi_\beta^H(\bar{z}|s)} \right)^{\frac{1}{4}} \sqrt{\frac{1}{2}\Delta_c} \right] \tag{63}$$

$$= \kappa_1 \tag{64}$$

We get $\kappa = \kappa_1 + \kappa_2$. We apply $\mathcal{O}$ to get the notation in the theorem. $\qquad \square$

## C  EXPERIMENT DETAILS

### C.1  OPAL EXPERIMENT DETAILS

**Encoder** The encoder $q_\phi(z|\tau)$ takes in state-action trajectory $\tau$ of length $c$. It first passes the individual states through a fully connected network with 2 hidden layers of size $H$ and ReLU activation. Then it concatenates the proccessed states with actions and passes it through a bidirectional GRU with hidden dimension of $H$ and 4 GRU layers. It projects the output of GRU to mean and log standard deviation of the latent vector through linear layers.

**Prior** The prior $\rho_\omega(z|s)$ takes in the current state $s$ and passes it through a fully connected network with 2 hidden layers of size $H$ and ReLU activation. It then projects the output of the hidden layers to mean and log standard deviation of the latent vector through linear layers.

**Primitive Policy** The primitive (i.e. decoder) $\pi_\theta(a|s,z)$ has same architecture as the Prior but it takes in state and latent vector and produces mean and log standard deviation of the action. For kitchen environments, we use an autoregressive primitive policy with same architecture as used by EMAQ (Ghasemipour et al., 2020).

We use $H = 200$ for antmaze environments and $H = 256$ for kitchen environments. In both cases, OPAL was trained for 100 epochs with a fixed learning rate of $1e-3$, $\beta = 0.1$ (Lynch et al., 2020), Adam optimizer (Kingma & Ba, 2014) and a batch size of 50.

### C.2  TASK POLICY ARCHITECTURE

In all environments, for task policy, we used a fully connected network with 3 hidden layers of size 256 and ReLU activation. It then projects the output of the hidden layers to mean and log standard deviation of the latent vector through linear layers.

| Environment | $\dim(\mathcal{Z}) = 4$ | $\dim(\mathcal{Z}) = 8$ | $\dim(\mathcal{Z}) = 16$ |
|---|---|---|---|
| antmaze medium (diverse) | $68.7 \pm 2.3$ | $81.1 \pm 3.1$ | $81.3 \pm 1.8$ |

Table 5: Average success rate (%) (over 4 seeds) of CQL+OPAL for different values of $\dim(\mathcal{Z})$. We fix $c = 10$.

### C.3 SAC HYPERPARAMETERS

We used SAC (Haarnoja et al., 2018) for online RL experiments in learning a task policy either in action space $\mathcal{A}$ or latent space $\mathcal{Z}$. For the discrete primitives extracted from DDCO (Krishnan et al., 2017), we used Double DQN Van Hasselt et al. (2015). We used the standard hyperparameters for SAC and Double DQN as provided in rlkit code base (`https://github.com/vitchyr/rlkit`) with both policy learning rate and q value learning rate as $3e - 4$.

### C.4 CQL HYPERPARAMETERS

We used CQL (Kumar et al., 2020b) for offline RL experiments in learning a task policy either in action space $\mathcal{A}$ or latent space $\mathcal{Z}$. We used the standard hyperparameters, as mentioned in Kumar et al. (2020b)., with minor differences. We used policy learning rate of $3e - 5$, q value learning rate of $3e - 4$, and primitive learning rate of $3e - 4$. For antmaze tasks, we used CQL($\mathcal{H}$) variant with $\tau = 5$ and learned $\alpha$. For kitchen tasks we used CQL($\rho$) variant with fixed $\alpha = 10$. In both cases, we ensured $\alpha$ never dropped below $0.001$.

## D CONNECTION BETWEEN OPAL AND VAE OBJECTIVES

We are given an undirected, unlabelled and diverse dataset $\mathcal{D}$ of sub-trajectories of length $c$. We would like to fit a sequential VAE model to $\mathcal{D}$ which maximizes

$$\max_{\theta} \mathbb{E}_{\tau \sim \mathcal{D}}[\log p_{\theta}(\tau|s_0)] \tag{65}$$

where $s_0$ is initial state of the sub-trajectory. Let's consider

$$\log p_{\theta}(\tau|s_0) = \log \int p_{\theta}(\tau, z|s_0)dz = \log \int \frac{p_{\theta}(\tau, z|s_0)q_{\phi}(z|\tau)}{q_{\phi}(z|\tau)}dz \tag{66}$$

(using Jensen's inequality)

$$\geq \int q_{\phi}(z|\tau)[\log p_{\theta}(\tau, z|s_0) - \log q_{\phi}(z|\tau)]dz = \mathbb{E}_{z \sim q_{\phi}(z|\tau)}\left[\log p_{\theta}(\tau|z, s_0) - \log \frac{q_{\phi}(z|\tau)}{p_{\theta}(z|s_0)}\right] \tag{67}$$

Using the above equation, we have the following lower-bound for our objective function

$$\max_{\theta} \mathbb{E}_{\tau \sim \mathcal{D}}[\log p_{\theta}(\tau|s_0)] \geq \max_{\theta, \phi} \mathbb{E}_{\tau \sim \bar{D}, z \sim q_{\phi}(z|\tau)}\left[\log p_{\theta}(\tau|z, s_0) - \log \frac{q_{\phi}(z|\tau)}{p_{\theta}(z|s_0)}\right] \tag{68}$$

$$= \max_{\theta, \phi} \mathbb{E}_{\tau \sim \mathcal{D}, z \sim q_{\phi}(z|\tau)}[\log p_{\theta}(\tau|z, s_0)] - \mathrm{D_{KL}}(q_{\phi}(z|\tau)||p_{\theta}(z|s_0)) \tag{69}$$

We separate the parameters of decoder from prior and hence write $p_{\theta}(z|s_0) = \rho_{\omega}(z|s_0)$. We can expand $\log p_{\theta}(\tau|z, s_0) = \sum_{t=1}^{c-1} \log \mathcal{P}(s_t|s_{t-1}, a_{t-1}) + \sum_{t=0}^{c-1} \log \pi_{\theta}(a_t|s_t, z)$. Since $\mathcal{P}$ is fixed it can be removed from the objective function. Therefore, we can write the objective function as

$$\max_{\theta, \phi} \mathbb{E}_{\tau \sim \mathcal{D}, z \sim q_{\phi}(z|\tau)}\left[\sum_{t=0}^{c-1} \log \pi_{\theta}(a_t|s_t, z)\right] - \beta \mathrm{D_{KL}}(q_{\phi}(z|\tau)||\rho_{\omega}(z|s_0)) \tag{70}$$

where $\beta = 1$. This is similar to the autoencoding loss function we described in section 4.

## E ABLATION STUDIES

As shown in Table 5, we experimented with different choices of $\dim(\mathcal{Z})$ on antmaze-medium (diverse). Using the hyperparameters from Nachum et al. (2018a), we fixed $c = 10$. While

$\dim(\mathcal{Z}) = 8, 16$ gave similar performances, $\dim(\mathcal{Z}) = 4$ performed slightly worse. Therefore, we selected $\dim(\mathcal{Z}) = 8$ for our final model as it was simpler.

**Temporal abstraction actually helps:** To empirically verify that the gain in performance was due to temporal abstraction and not better action space learned through latent space, we tried $c = 1$ ($\dim(\mathcal{Z}) = 8$) and found the performance to be similar to that of CQL (i.e. $55.3 \pm 3.8$) thereby empirically supporting the theoretical benefits of temporal abstraction.

We found $\dim(\mathcal{Z}) = 8$ and $c = 10$ to work well with other environments as well. However, we acknowledge that the performance of CQL+OPAL can be further improved by carefully choosing better hyperparameters for each environment or by using other offline hyperparameter selection methods for offline RL, which is a subject of future work.

## F ALTERNATIVE METHODS FOR EXTRACTING PRIMITIVES FROM OFFLINE DATA

We describe alternative methods for extracting a primitive policy from offline data. These methods are offline variants of CARML (Jabri et al., 2019) and DADS (Sharma et al., 2019). We tried these techniques in an early phase of our project and used the environment antmaze-medium (diverse) to evaluate these methods.

Let's consider an offline undirected, unlabelled and diverse dataset $\mathcal{D} = \{(s_t^i, a_t^i)_{t=0}^{c-1}\}_{i=1}^N$. Let $\tau = (s_t)_{t=0}^{c-1}$ represent the state trajectory. To extract primitives, we first cluster the trajectories by maximizing the mutual information between the state trajectory $\tau$ and latent variable $z$ (indicating cluster index) with respect to the parameters of joint distribution $p_{\phi,\omega}(\tau, z) = p_\omega(z)p_\phi(\tau|z)$. For now, we consider $p_\omega(z) = \text{Cat}(p_1, \ldots, p_k)$ (i.e. discrete latent variables sampled from a Categorical distribution) and represent $z$ as one-hot vector of dimension $k$. The choice of $p_\omega(z)$ is consistent with the choices made in Jabri et al. (2019) and Sharma et al. (2019). Since $z$ is discrete, we can use Bayes rule to calculate $p_{\phi,\omega}(z|\tau)$ as

$$p_{\phi,\omega}(z|\tau) = \frac{p_\omega(z)p_\phi(\tau|z)}{\sum_{i=1}^k p_\omega(z_i)p_\phi(\tau|z_i)}. \tag{71}$$

Our objective function becomes

$$\max_{\phi,\omega} I(\tau; z) = \max_{\phi,\omega} \mathbb{E}_{\tau \sim \mathcal{D}, z \sim p_{\phi,\omega}(z|\tau)} \left[ \log \frac{p_\phi(\tau|z)}{p(\tau)} \right]. \tag{72}$$

Offline CARML and offline DADS differ only in how they model $p_\phi(\tau|z)$:

- **Offline CARML** We model $p_\phi(\tau|z) = \prod_{t=0}^{c-1} p_\phi(s_t|z)$ and hence, $\log p_\phi(\tau|z) = \sum_{t=0}^{c-1} \log p_\phi(s_t|z)$.

- **Offline DADS** We model $p_\phi(\tau|z) = p(s_0) \prod_{t=1}^{c-1} p_\phi(s_t|s_{t-1}, z)$ and hence, $\log p_\phi(\tau|z) = \log p(s_0) + \sum_{t=0}^{c-1} \log p_\phi(s_t|s_{t-1}, z)$. Here, we only model $p_\phi(s_t|s_{t-1}, z)$ and not $p(s_0)$. Since, $\log$ is additive in nature, $p(s_0)$ will be ignored while calculating gradient.

To optimize equation 72, we use *Algorithm 2* from Jabri et al. (2019). Once we have clustered the state trajectories $\tau$ with labels $z$ by maximizing $I(\tau; z)$, we can use behavioral cloning (BC) to learn $\pi_\theta(a|s, z)$.

Finally, we use $p_{\phi,\omega}(z|\tau)$ to label the reward-labelled data $\mathcal{D}^r = \{(s_t^i, a_t^i, r_t^i)_{t=0}^{c-1}\}_{i=1}^N$ with latents, and transform it into $\mathcal{D}_{\text{hi}}^r = \{(s_0^i, z_i, \sum_{t=0}^{c-1} \gamma^i r_t^i, s_c)_{i=1}^N\}$. The task policy $\pi_\psi$ is trained on $\mathcal{D}_{\text{hi}}^r$ using Conservative Q Learning (CQL) (Kumar et al., 2020b). Since the primitive policy $\pi_\theta$ is trained after $p_{\phi,\omega}(z|\tau)$ is fully trained, it doesn't need any additional finetuning.

### F.1 RESULTS

Using the hyperparameters from Nachum et al. (2018a), we used $c = 10$. We experimented with different values of $k = 5, 10, 20$ and found that $k = 10, 20$ works the best (see Table 6 for more

| Models | $k = 5$ | $k = 10$ | $k = 20$ |
|---|---|---|---|
| CQL+Offline DADS | $31.4 \pm 5.7$ | $59.1 \pm 3.1$ | $59.6 \pm 2.9$ |
| CQL+Offline CARML | $13.3 \pm 4.7$ | $15.1 \pm 2.6$ | $14.9 \pm 3.8$ |

Table 6: Average success rate on antmaze medium (diverse) (%) (over 4 seeds) of CQL combined with offline DADS and offline CARML for different values of $k$.

| Environment | CQL | CQL+OPAL | CQL+ DADS | CQL+ CARML |
|---|---|---|---|---|
| success rate | $53.7 \pm 6.1$ | $\mathbf{81.1 \pm 3.1}$ | $59.1 \pm 3.1$ | $15.1 \pm 2.6$ |
| cumulative dense reward | $-12138.6 \pm 720.3$ | $\mathbf{-7795.7 \pm 535.4}$ | $-11184.7 \pm 610.1$ | $-13387.3 \pm 710$ |
| cumulative dense reward (last 5 steps) | $-45.1 \pm 9.2$ | $\mathbf{-7.8 \pm 4.6}$ | $-33.1 \pm 8.1$ | $-51.6 \pm 5.6$ |

Table 7: Average success rate (%), cumulative dense reward, and cumulative dense reward (last 5 steps) (over 4 seeds) of CQL combined with different offline skill discovery methods on antmaze medium (diverse). For CQL + (Offline) DADS and CQL + (Offline) CARML, we use $k = 10$. Note that CQL+OPAL outperforms both other methods for unsupervised skill discovery on all of these different evaluation metrics.

details). We went with $k = 10$ as our final model since it's simpler. Offline CARML effectively uses only 6 skills as the other 4 skills had $p_\omega(z) = 0$. Offline DADS uses all the skills. The results are described in Table 7. In addition to calculating the average success rate, we also calculate the average cumulative dense rewards for entire trajectory and the average cumulative dense rewards for the last 5 time steps. Here, the dense reward is negative $l_2$ distance to the goal. The resulting trajectory clusters (using a subset of the dataset) from discrete skills are also visualized in Figure 4 where different colors represent different clusters.

Since offline CARML treats the states in the trajectory conditionally independent of each other given $z$, the clustering mainly focuses on the spatial location. Therefore, offline CARML isn't able to separate out different control modes starting around the same spatial locations which explains its poor performance when combined with CQL. As we can see from Figure 5, offline CARML is able to make progress towards the goal, but gets stuck along the way due to poor separation of control modes. On the other hand, offline DADS treats the state transitions in the trajectory conditionally independent of each other given $z$ and thus clusters trajectories with similar state transitions together. This allows it to more effectively separate out the control modes. Therefore, CQL+offline DADS slightly improves upon CQL but is still limited by discrete number of skills. Furthermore, increasing the number of skills from 10 to 20 gives similar performance. Moreover, in these methods, it's intractable to use continuous skill space since we use Bayes rule to calculate $p_{\phi,\omega}(z|\tau)$. Therefore, we decided to switch to learning a $\beta$-VAE (Higgins et al., 2016) style generative model with continuous skill space i.e. OPAL.

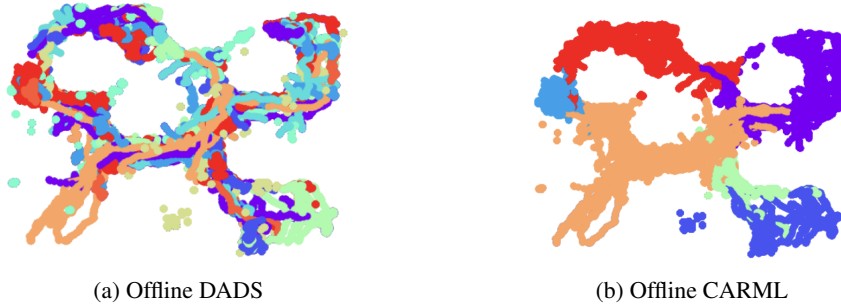

(a) Offline DADS                    (b) Offline CARML

Figure 4: Visualization of (a subset of) dataset trajectories colored according to their assigned cluster using (a) Offline DADS and (b) Offline CARML. We use $k = 10$.

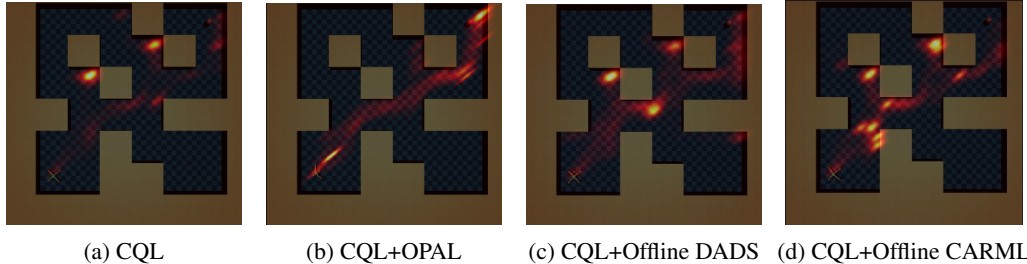

    (a) CQL         (b) CQL+OPAL      (c) CQL+Offline DADS  (d) CQL+Offline CARML

Figure 5: State visitation heatmaps for antmaze medium policies learned using (1) CQL, (2) CQL+OPAL, (2) CQL+Offline DADS and (4) CQL+Offline CARML. Note that while offline CARML and offline DADS get stuck at various corners of the maze, OPAL is able to find its path through the maze to the goal location on the top right.

### F.2 TRAINING DETAILS

For clustering by optimizing equation 72, both offline CARML and offline DADS only considers the global x-y pose of the ant and ignores other dimensions of the state space. These methods fail to work when considering the full state space.

**Offline CARML** $p_\phi(s|z)$ takes in the latent one-hot vector $z$ and passes it through a fully connected network with 2 hidden layers of size $H = 200$ and ReLU activation. It then projects the output of the hidden layers to mean and log standard deviation of the reduced state $s$ (only global x-y pose considered) through linear layers.

**Offline DADS** $p_\phi(s_t|s_{t-1}, z)$ has the same architecture as $p_\phi(s|z)$ but also takes in the reduced state from the previous timestep.

**Primitive Policy** The primitive policy $\pi_\theta(a|s, z)$ takes in the current state $s$ and latent one-hot vector $z$ and passes it through a fully connected network with 2 hidden layers of size $H = 200$ and ReLU activation. It then projects the output of the hidden layers to mean and log standard deviation of action through linear layers.

We perform the clustering for 25 epochs with a fixed learning rate of $1e - 3$, Adam optimizer (Kingma & Ba, 2014) and a batch size of 50 using the *Algorithm 2* from Jabri et al. (2019).

**Task Policy** For task policy $\pi_\psi(s)$, we used a fully connected network with 3 hidden layers of size 256 and ReLU activation. It then projects the output of the hidden layers to the logits (corresponding to the components of discrete latent space) through linear layers.

**CQL Hyperparameters** We used the standard hyperparameters for CQL($\mathcal{H}$) with discrete action space, as mentioned in Kumar et al. (2020b).

