# OpenReview forum: "OPAL: Offline Primitive Discovery for Accelerating Offline Reinforcement Learning"
_ICLR.cc/2021/Conference — ICLR 2021 Poster_

### Official Review · AnonReviewer1 · 2020-10-22
**Interesting research question but motivation of the method unclear**

**Rating:** 6
**Confidence:** 3

**Review:**

In the RL setting, this paper tackles the case where an agent may have access to large amounts of offline experience data. The objective of the work is to find an effective way to leverage this data for finding temporally extended primitive behaviors. The paper provides results that show how performing offline primitive learning can be leveraged for improving few-shot imitation learning as well as exploration and transfer on a variety of benchmark domains.

The paper tackles an important question in reinforcement learning: learning temporally extended primitive behaviors from off-policy data is a very relevant question. However, I found the motivation for the approach quite vague as well as different elements that require clarification (see below) and because of this, I can't recommend acceptance.

Motivation for the approach:
- if the downstream policy has to stay close to the offline data distribution, it seems to me that if the offline data distribution is obtained with a bad policy, this can not lead to interesting decision-making.
- can you explain why equation 2 "motivates better generalization"?

Other concerns:
- Why can gamma not have a value of 0 (first paragraph preliminaries)
- What does it mean "To capture multi-modality in a dataset" ? (second paragraph preliminaries)
- Figure 1:  What does the (81.1%) and (70.3%) refer to?
- Table 4: why is SAC given without standard deviation?

Some text improvements:
- "we focus on focus on"

Other comment:
- The number of seeds (4) should ideally be increased.

---

> ### Author Response · Authors · 2020-11-19
> **Author Response (Part 1/2): Clarifications about motivation, questions in the paper**
>
> We thank the reviewer for their constructive feedback. We have updated the motivation -- both in the paper (Section 1) as well as in a shared response to the reviewers, here, on openreview. We will address the reviewer’s concern as follows:
>
> **“The objective of the work is to find an effective way to leverage this data for finding temporally extended primitive behaviors.”**
>
> **RESPONSE:** We would say that the motivation is to use effective (and simple) ways to accelerate offline RL rather than finding an effective way to leverage offline data for finding temporally extended primitive behaviors. We have now made this very explicit in Section 1 of the paper.  One such effective way is to first extract temporally extended primitives from offline data and then leverage those primitives to accelerate offline RL algorithms. While this might seem intuitive at first, it is actually not so intuitive, given the fact that prior works that performed controlled empirical analysis on the benefits of hierarchical RL (Nachum et al, 2019) find that hierarchies work better than standard online RL because of improved exploration and there is absolutely no exploration to be done in an offline RL setting. To the best of our knowledge, ours is the first work to theoretically justify and experimentally verify the benefits of primitive learning in offline settings, showing that hierarchies can provide temporal abstraction that allows us to reduce the effect of compounding errors issue in offline RL. This is the reason behind our motivation towards demonstrating the efficacy of learning primitives in an offline RL perspective. Please see the shared response as well for clarifications on motivations of this project.
>
> **“if the downstream policy has to stay close to the offline data distribution, it seems to me that if the offline data distribution is obtained with a bad policy, this can not lead to interesting decision-making”**
>
> **RESPONSE:** We emphasize that our use of OPAL restricts the learned policy to “stay close” to the offline data distribution in the sense of restricting the learned policy to be within the support of the offline data. This is a much weaker notion than many existing offline RL algorithms that use explicit KL or other divergence regularizers to limit the deviation of the learned policy from the data-generating policy. To see why this is the case, note that by decomposing the policy into primitives, and by performing RL over the latent variables $z$, we are allowed to select actions that are within the support of the behavior policy but are not the mean of the behavior policy distribution, which is what a KL-divergence style constraint typically used in offline RL would do.  While the support-based notion we use still restricts the expressivity of the learned policy (as specifically analyzed by our theoretical statements), this is arguably unavoidable; for example, see a concurrent submission deriving hardness results of learning policies based on offline data that is far from the optimal policy https://openreview.net/forum?id=30EvkP2aQLD
>
> **“can you explain why equation 2 "motivates better generalization"?”**
>
> **RESPONSE:** Without equation 2, the autoencoding may encourage the encoder to assign a unique z to every sub trajectory. As argued in beta-VAE paper (Higgins et al 2017), the constraint in equation 2 allows us to control the informational capacity of the latent information bottleneck, which helps with generalization
>
> Equation 2 regularizes the latent space of z and forces the distribution of z to depend on the initial state of the sub-trajectory rather than the entire sub-trajectory.
>
> **“Why can gamma not have a value of 0 (first paragraph preliminaries)”**
>
> **RESPONSE:** We are interested in long-horizon sequential decision making problems that are typically the focus of reinforcement learning algorithms. In this setting if $\gamma$ is set to 0, the episode length reduces to a single time-step and the decision making problem effectively loses its sequential structure. In this work, we are interested in tasks with long horizons where we can leverage temporal abstraction. Therefore, we assume $\gamma$ > 0, and we have clarified this detail in the paper.
>
> **“What does it mean "To capture multi-modality in a dataset" ? (second paragraph preliminaries)”**
>
> **RESPONSE:** In this work, we are focusing on multi-modal datasets (i.e. the distribution of actions $a$ at a given observation $s$ in the dataset can be multimodal -- in simpler words, from the same state s, the agent can perform 2 or more different behaviors). To model the process of dataset $\mathcal{D}$ generation, we assume that an initial state is sampled first and then a policy is sampled from a distribution of policies which is then executed for $c$ timesteps -- this data generation procedure formalizes the notion of multi-modality.

---

> > ### Author Response · Authors · 2020-11-19
> > **Author Response (Part 2/2): Other questions and references**
> >
> > **“Figure 1: What does the (81.1%) and (70.3%) refer to?”**
> >
> > **RESPONSE:** In this figure, we utilize the Antmaze medium and large datasets from the D4RL offline RL benchmark (Fu et al. 2020). In accordance with this benchmark evaluation procedure, we report the success rates of every method in Figure 1 and those numbers correspond to this success rate metric. 81.1% and 70.3% are the success rates of OPAL+CQL on antmaze medium and antmaze large respectively, which is also discussed in Table 1 along with a comparison to baselines. We have now removed those numbers from Figure 1 to prevent confusion. Additionally, for better visualizations, we added state visitation heatmaps for the learned offline policies in Figure 3.
> >
> > **“Table 4: why is SAC given without standard deviation?”**
> >
> > **RESPONSE:** Those numbers are taken directly from previous work (Yu et al (2019)), and they don’t include the standard deviation. But, we will run these results and update the paper in the final version as suggested by the reviewer.
> >
> > **“The number of seeds (4) should ideally be increased.”**
> >
> > **RESPONSE:** We will run more seeds for all our experiments and update the paper in the final version as suggested by the reviewer.
> >
> > **REFERENCES:**
> > 1) beta-VAE: Learning Basic Visual Concepts with a Constrained Variational Framework. Irina Higgins, Loic Matthey, Arka Pal, Christopher Burgess, Xavier Glorot, Matthew Botvinick, Shakir Mohamed, Alexander Lerchner
> > 2) D4RL: Datasets for Deep Data-Driven Reinforcement Learning. Justin Fu, Aviral Kumar, Ofir Nachum, George Tucker, Sergey Levine
> > 3) Meta-World: A Benchmark and Evaluation for Multi-Task and Meta Reinforcement Learning. Tianhe Yu, Deirdre Quillen, Zhanpeng He, Ryan Julian, Karol Hausman, Chelsea Finn, Sergey Levine
> > 4) Why Does Hierarchy (Sometimes) Work So Well in Reinforcement Learning? Ofir Nachum, Haoran Tang, Xingyu Lu, Shixiang Gu, Honglak Lee, Sergey Levine

---

> > ### Comment · AnonReviewer1 · 2020-11-23
> > **Answer to authors**
> >
> > Thanks for your clarifications. My main remaining concerns are the following:
> >
> > 1. The authors have answered to my comments "Equation 2 regularizes the latent space of z and forces the distribution of z to depend on the initial state of the sub-trajectory" : Now that you explain it this way, it seems a rather ad hoc way of regularizing. Since this seems a rather important element that motivates the whole algorithm, was this choice explained in depth in the paper? I only see the explanation mentioning "a KL constraint to encourage better generalization".
> >
> > 2. Concerning the multi-modality term, I'm not very convinced by the use of this word. Has that word been used in other works in the same way than the one you explain?

---

> > > ### Author Response · Authors · 2020-11-23
> > > **Thank you for the follow up! Addressing the remaining concerns**
> > >
> > > We thank the reviewer for their prompt response, and we are glad that most of their concerns are addressed. To address the remaining concerns, we have updated the paper to discuss the regularization in Equation 2 in Section 4.1 (now highlighted in violet)  and provide answers to specific questions below.
> > >
> > > **Equation 2…..was this choice explained in depth in the paper**
> > > We would like to point out that this choice of regularization of latent space distribution is inspired from prior works (Lynch et al. (2019), Kumar et al. (2019)). To be exact, Kumar et al. uses a cross-entropy loss than the complete KL divergence but also has prior which depends on initial state. As highlighted in section 4.1, having our prior be dependent on the initial state allows us the flexibility of having different sets of primitives active from different initial states. While this choice is inspired from previous work mentioned above, presumably other regularizers could work well too.
> > >
> > > **Concerning the multi-modality term, I'm not very convinced by the use of this word. Has that word been used in other works in the same way than the one you explain?**
> > > The term multi-modality has been used in the same way in Haarnoja et al (2017) (Section 2.2), Hausman et al (2017) (Section 4), Lynch et al (2019) (Section 3.1). We are happy to replace the term with any other term if the reviewer has any suggestions in the final version of this paper.
> > >
> > > We would appreciate it if the reviewer can let us know if these answers address their concerns. We would be happy to address any more concerns that the reviewer has.
> > >
> > >
> > > **References**
> > >
> > > (1) Reinforcement Learning with Deep Energy-Based Policies. Tuomas Haarnoja, Haoran Tang, Pieter Abbeel, Sergey Levine
> > >
> > > (2) Multi-Modal Imitation Learning from Unstructured Demonstrations using Generative Adversarial Nets. Karol Hausman, Yevgen Chebotar, Stefan Schaal, Gaurav Sukhatme, Joseph Lim
> > >
> > > (3) Learning Latent Plans from Play. Corey Lynch, Mohi Khansari, Ted Xiao, Vikash Kumar, Jonathan Tompson, Sergey Levine, Pierre Sermanet
> > >
> > > (4) Learning Navigation Subroutines from Egocentric Videos. Ashish Kumar, Saurabh Gupta, Jitendra Malik

---

> > > > ### Comment · AnonReviewer1 · 2020-11-24
> > > > **Clarifications not perfectly clear**
> > > >
> > > > Thanks for the tentative of clarifications here and in the paper. Concerning point 1, the setting does not seem to be directly related to the references that you are mentioning here and that you have added in the paper. The justification of that regularization is key to the paper and I don't have enough explanation/context from the paper to increase my score. Some toy examples or intuitive explanations of the different choices might help improve the clarity of the paper.
> > > >
> > > > Just as a side note concerning the second point, I looked up the references but it didn't help in clarifying how this term would be suited for characterizing that "from the same state s, the agent can perform 2 or more different behaviors".
> > > >
> > > > Overall, I feel that the key take aways from the paper are unclear and I therefore can't increase my score of marginally below acceptance (5).

---

> > > > > ### Author Response · Authors · 2020-11-25
> > > > > **Thanks for the response! Clarifying in more detail.**
> > > > >
> > > > > We thank the reviewer for their prompt response. We apologize that our clarifications were not clear. We have updated the paper (in violet color, top of Page 5) to address the concern about KL-constraint in detail, and we summarize these points below.
> > > > >
> > > > > **Clarification for the KL-constraint (Equation 2):** We have now edited Section 4.1 to discuss the KL-constraint and the intuitive rationale behind in more detail. This KL constraint regularizes the distribution of primitive or latent variables and allows us the ability to have different sets of primitives active given different initial states. To briefly summarize its relation to prior work, example, Lynch et al. 2019, we point the reviewer to Equation 2 and the "``Plan prior matching" section of Lynch et al. 2019, which uses a similar KL constraint between the distribution over latents given the start state and the goal state (we do not have a goal state in our setting, and so $\rho_\omega(z|s_0)$ only uses the start state $s_0$ ) and the distribution over latents given the trajectory (which in our cases is the encoder, $q_\phi(z|\tau)$.)
> > > > >
> > > > > We would also like to clarify that this *Equation 2 is not the primary component of our work*. Equation 1 presents an autoencoding loss, and our theoretical results showing the benefits of temporal abstraction focus on the objective in Eq 1 as well as specific characteristics of the data distribution, rather than the KL regularizer in Eq 2. As we clarify, we view the KL regularization as a way to encourage regularization and the availability of different primitives at a state in practice; while this choice is inspired from previous work mentioned above, presumably other regularizers could work well too.
> > > > >
> > > > > **Multi-modality:** We have removed all occurrences of this word from the paper, and we have highlighted this change in violet in Section 3, where "multi-modal" was previously used. However, we would like to note that Lynch et al. 2019, in Section 3.1, uses the term **“Multimodality problem”** and discusses: “A challenge in self-supervising control on top of play is that in general, **there are many valid high-level behaviors that might connect the same $(s_c, s_g)$ pair**. This presents multiple counteracting action label trajectories, which can impede learning. Therefore, policies must be expressive enough to model all possible high-level behaviors that lead to the same goal outcome.”  So we used this word in the first place. But we have removed it now from our paper completely as suggested by the reviewer.
> > > > >
> > > > > >Overall, I feel that the key take aways from the paper are unclear and I therefore can't increase my score of marginally below acceptance (5).
> > > > >
> > > > > We believe that our paper both theoretically and empirically shows that temporal abstraction via primitive learning can aid offline RL by reducing the compounding errors issue due to distributional shift, which is surprising in the right of the fact that hierarchy helps in online RL due to exploration reasons (Nachum et al. 2019). In practice, we utilize a simple scheme for learning primitives inspired from a beta-VAE in conjunction with an offline RL algorithm.
> > > > >
> > > > >
> > > > > We would appreciate it if the reviewer can let us know if these changes address their concerns. We are happy to also clarify any other concerns that the reviewer has about the takeaways of the paper if they can elaborate on which part seems unclear.

---

> ### Author Response · Authors · 2020-11-21
> **Discussion**
>
> Dear reviewer,
>
> Please let us know if our response below and the shared response about motivation addresses the concerns raised in your review. We will be happy to clarify these or other concerns more.

---

### Official Review · AnonReviewer2 · 2020-10-28
**Good paper with (potentially) practical values for real-world robotic tasks**

**Rating:** 7
**Confidence:** 3

**Review:**

## OPAL: Offline Primitive Discovery for Accelerating Offline Reinforcement Learning
### Summary
The authors present OPAL, an offline reinforcement learning approach that distills useful common behaviors from offline transition data. They leverage a variational GRU to encode trajectories into a latent space of primitive policies. These learned primitives can later on be transferred to other tasks by learning a high-level controller over the extracted primitive latent space. Their experimental results show that OPAL compares favorably against other SOTA methods in offline, online and few-shot reinforcement learning settings.

Overall, the paper is clearly written, and the approach is potentially practical for real-world applications, due to its good performance on both offline and few-shot adaptation settings.

### Strength
- The proposed method can leverage large unlabeled data without pre-defined reward functions.
- The property of OPAL is mathematically analyzed.
- OPAL can potentially benefit from the advancement of offline RL algorithms by simply replacing CQL.
- Can be adapted to a new task through few-shot imitation learning.

### Weakness
- Training details are missing for the OPAL encoder.
- Not exactly a weakness, but it would be nice to see if the method works in real-world settings (real-world on/offline data).

---

> ### Author Response · Authors · 2020-11-19
> **Author Response**
>
> We thank the reviewer for their constructive feedback. We will address the reviewer’s concern as follows:
>
> **“Training details are missing for the OPAL encoder”**
>
> **RESPONSE:** The encoder is trained through equation (1) and (2) in the paper. As mentioned in the paper in Section 4.1, we combine (1) and (2) and convert them into a $\\beta$ VAE formulation. Therefore, the overall objective function becomes
>
> $<\\hat{E}_{\\tau \\sim D, z \\sim q_{\\phi}(z|\\tau)}[-\\sum_{t=0}^{c-1} \\log \\pi_{\\theta}(a_t|s_t,z)] + \\beta \\hat{E}_{\\tau \\sim D}[D_{KL}(q_{\\phi}(z|\\tau)||\\rho_{\\omega}(z|s_0))]>$
>
> The encoder $q_\\phi$ is trained using the above objective function through reparameterization and backpropagation.
>
> The training details for OPAL (which includes all the architectural details and hyperparameter choices) is provided in Appendix C.
>
> **“Not exactly a weakness, but it would be nice to see if the method works in real-world settings (real-world on/offline data)”**
>
> **RESPONSE:** We planned to do some simple real world maze navigation experiments using D’kitty robot from the ROBEL benchmark (Ahn et al, 2019). However, access to the lab has been limited due to COVID. However, we aim to add this experiment on a real-world task for the final version of this paper.
>
> **REFERENCES:**
> 1) ROBEL: RObotics BEnchmarks for Learning -- Low-Cost, Robust and Reproducible. Michael Ahn, Henry Zhu, Kristian Hartikainen, Hugo Ponte, Abhishek Gupta, Sergey Levine, VIKASH KUMAR

---

> > ### Comment · AnonReviewer2 · 2020-11-21
> > **Thanks for the updates**
> >
> > Thanks for the updates.
> >
> > After reading other reviewers' comments and your responses, my concerns are addressed, and currently I do not have any problem to ask.
> >
> > As of now, I intend to stick with my original evaluation, but I will keep track of the discussions.

---

### Official Review · AnonReviewer3 · 2020-10-29
**While interesting,  baselines are weak and OPAL does not directly address problems in offline RL**

**Rating:** 6
**Confidence:** 4

**Review:**

Summary
-------

To best leverage diverse datasets of logged experience, the authors
propose to extract a space of primitive behaviors in a continuous space,
and to use these for downstream learning. The primitive behaviors are
learned through a VAE loss, and CQL is applied to learn a policy over
the primitives. The authors claim that this approach to offline RL
avoids known distribution shift and allows for temporal abstraction. The
method is also applied to few-shot imitation learning, exploration and
transfer to online RL.

Decision
--------

Despite this paper being an interesting read, I feel that my concerns
about the experiments lead me to not be confident in the proposed
approach. As such, my preliminary rating for the paper is "Okay but not
good enough". The baselines chosen for the experiments do not seem
representative of the problem being addressed. This also leads into the
motivation, where OPAL is motivated from the offline RL perspective but
does not explicitly mitigate the issues in offline RL. Only the
theoretical results investigate the effect of a fixed replay-buffer, but
even these claims are framed in terms of what $\mathcal{D}$ should be to
ensure downstream RL.

Originality
-----------

The proposed approach seems limited in its novelty, combining approaches
from skill-discovery, hierarchical RL and somewhat from offline RL. The
theory section is interesting, however the technical arguments seem
similar to that of Nachum et al. (2018).

Quality and Clarity
-------------------

The overall quality of the paper is very high. The writing is clear, and
the theoretical arguments are put into an easily understandable context.

Strengths
---------

-   OPAL leverages techniques from many areas of machine learning:
    unsupervised learning, hierarchical RL and offline RL. This is an
    interesting combination, and deserves to be investigated. It seems
    like the workhorse of OPAL is the unsupervised learning component
    however, and perhaps this should be emphasized and investigated
    independently.
-   The theoretical section is very clear and well contextualized. The
    claims in the paper seem correct, and they do provide much needed
    validation for hierarchical RL. The assumption of an optimal
    high-level controller seems strong, but the analysis is nonetheless
    interesting.

Weaknesses
----------

-   Despite Figure 2, OPAL is difficult to disentangle with many moving
    parts. It doesn't help that there is no discussion of how
    hyperparameters were chosen for each component, or an ablation study
    to investigate different hierarchical RL approaches, fine-tuning,
    offline RL algorithms, or VAE losses.
-   The proposed method is not specifically designed to leverage
    anything particular in offline RL. As you show, it can be applied to
    online RL. However, why is this motivated from the offline
    perspective? OPAL itself does not seem to mitigate distributional
    shift.
-   It would be helpful to directly compare to other skill-discovery
    methods that have been applied to online RL but can be adapted to
    offline RL. For example, what prevents the work by Campos et
    al. (2020) and the baselines therein to be applied in conjunction
    with an offline RL algorithm? The baselines for the experiments do
    not seem representative of the problem you are addressing. You use
    standard offline RL algorithms, yet claim that the temporal
    abstraction of skill discovery is crucial to the results. As such,
    you should compare OPAL to other skill-discovery algorithms that can
    be combined with offline RL.

Detailed Comments
-----------------

-   Section 4.1: " Prior: ρω(z|s0) tries to predict the encoded
    distribution of the sub-trajectory.." Prior doesn't seem like the
    right word here, since it is being learned.
-   Section 4.2: In what way is the task-specific policy $\pi_\psi$
    different from the learned prior in Section 4.1? Both are
    distributions over latent $z$ conditioned on a state. While the
    prior is designed to only be conditioned on the initial state, this
    is still quite similar to the high-level policy $\pi_\psi$.
-   Section 4.2: "\[to\] ensure that the c-step transitions remain
    consistent … with the labelled latent action $z_i$" Why is this
    necessary, should CQL ensure consistency at the action-level, while
    the latent actions are consistent by design?
-   Corollary 4.1.1: How would $\epsilon$ actually approach
    $\mathcal{H}_c$ in the algorithm? The condition is never explicitly
    enforced because you formulate it as a constrained optimization
    problem and you are never able to change $\mathcal{H}_c$ since it is
    a constant.
-   Section 5.1, baselines: shouldn't baselines be compared to different
    unsupervised skill discovery algorithms, paired with offline RL
    algorithms? BEAR and EMAQ are offline RL algorithms without any
    temporal abstraction. And as you say in the results section with an
    ablation study, this temporal abstraction is crucial.
-   Section 5.1, results: An ablation study is discussed but I cannot
    find the corresponding results table/figure in the paper.
-   Section 5.2, Table 3: With so many 0.0's, this leads me to believe
    that the baselines are quite weak, or the problem is too hard for
    the baselines. How were the baselines chosen. For example, why was
    DDCO paired with DDQN instead of SAC?

Minor Comments
--------------

-   Section 5.2 mis-capitalized We "in this setting We use the Antmaze
    environ-"
-   Section 5.3, Table 4: why are there no standard errors for SAC?

Post Rebuttal
--------------
After discussion with the authors, I have decided to increase my score to a 6. The authors have addressed many of my concerns with respect to motivation, theoretical analysis and empirical evidence. As it stands, I still think OPAL is hampered by the many "moving parts" involved. The theoretical analysis and empirical evidence suggests a very effective approach however, and should lead to further work combining variational sequence encoding techniques for primitive extraction in RL. My decision would be to "accept the paper if there is room".

References
--------------
Campos, Víctor, Alexander Trott, Caiming Xiong, Richard Socher, Xavier Giro-i-Nieto, and Jordi Torres. 2020. “Explore, Discover and Learn: Unsupervised Discovery of State-Covering Skills.” *arXiv:2002.03647*.
<http://arxiv.org/abs/2002.03647v4>.

Nachum, Ofir, Shixiang Gu, Honglak Lee, and Sergey Levine. 2018. “Near-Optimal Representation Learning for Hierarchical Reinforcement Learning.” *arXiv:1810.01257*. <http://arxiv.org/abs/1810.01257v2>.

---

> ### Author Response · Authors · 2020-11-19
> **Author Response (Part 4/5): Other Concerns (Assumptions in Section 4, OPAL and Distribution Shift, Section 5)**
>
> **“OPAL itself does not seem to mitigate distributional shift”**
>
> **RESPONSE:** Intuitively OPAL partly mitigates distributional shift, by making the task policy sample primitives, which are restricted to be close to offline data distribution. Formally, OPAL significantly slows down the amount of error accumulation (which is caused due to distributional shift and sampling error) in offline RL. As is evident from the bound for offline policy performance in Theorem 4.2, the curse of distributional shift is significantly reduced when OPAL is used:  our bound depends on the size of the latent space and error is compounded by a smaller factor over the horizon (due to temporal abstraction) as compared to the standard RL counterpart of this bound which is affected by the size of the entire action space for each and every timestep in the horizon. This indicates that OPAL does help prevent distributional shift and sampling error.
>
> “Section 4.2: In what way is the task-specific policy $\pi_\psi$ different from the learned prior in Section 4.1? Both are distributions over latent  conditioned on a state. While the prior is designed to only be conditioned on the initial state, this is still quite similar to the high-level policy $\pi_\psi$ .”
>
> **RESPONSE:** The learned prior is learned using an autoencoding loss without knowledge of task (i.e reward) and tries to capture all the relevant skills from a state. On the other hand, the task specific policy is finetuned on the task in the RL setting, and so it focuses on skills relevant to maximizing the reward function
>
> **“ The assumption of an optimal high-level controller seems strong, but the analysis is nonetheless interesting.”**
>
> **RESPONSE:** Our analysis in Theorem 4.2 does not assume access to an optimal high-level controller. More specifically, Theorem 4.1 shows the existence of a nearly optimal high-level controller and builds the relationship between how the (sub) optimality of the high level controller depends on final autoencoding loss, dimension of the primitives and length of the temporally extended primitives ($c$). Once we have shown the existence of a near optimal high level controller, we show how to convert this into an offline RL result that learns the optimal controller in Theorem 4.2. Our conclusion is that using a high-level controller makes the offline RL problem easier due to reduced task horizon and how it asymptotically improves the performance of offline RL.
>
> **“however the technical arguments seem similar to that of Nachum et al. (2018)”**
>
> **RESPONSE:** We respectfully disagree with the statement. While Theorem 4.1 tries to argue the existence of a near optimal high level controller, the underlying assumption it makes and the subsequent proof techniques is very different from that of Nachum et al. (2018). Namely, Nachum et al. (2018) relies on the use of supremums of states in the MDP to provide a performance bound, and this limits its connection to practical algorithms. On the other hand, we prove the existence of a near optimal high level controller when underlying primitives are learned from a *stochastically sampled* offline dataset. Crucially, our derivations illuminate what properties the dataset should possess to ensure low suboptimality, and this is a key contribution of our work.
>
> Moreover, Theorem 4.2 is completely independent from Nachum et al (2018) and shows how temporally extended primitives asymptotically improve the performance of an underlying conservative offline RL method.
>
> **“Corollary 4.1.1: How would  $\epsilon$ approach $-\mathcal{H}_c$ in the algorithm? The condition is never explicitly enforced because you formulate it as a constrained optimization problem and you are never able to change  since it’s a constant.”**
>
> **RESPONSE:** While $\epsilon_c$ would decrease as a result of optimization, there’s no guarantee it would approach $-\mathcal{H}_c$. We have updated the statement in the paper.
>
> **“Section 5.3, Table 4: why are there no standard errors for SAC?”**
>
> **RESPONSE:** Those numbers are taken directly from previous work (Yu et al (2019)), and they don’t include the standard deviation. But, we will run these results and update the paper in the final version as suggested by the reviewer.
>
> **“Section 4.1: " Prior: ρω(z|s0) tries to predict the encoded distribution of the sub-trajectory.." Prior doesn't seem like the right word here, since it is being learned.”**
>
> **RESPONSE:** We borrowed this terminology from Kingma et al, 2013 which stated that the prior can be either fixed or learned. However, to make the paper less confusing, we have updated the paper to include the term ‘Primitive Predictor’ (as it predicts z from initial state) as well.

---

> > ### Author Response · Authors · 2020-11-19
> > **Author Response (Part 5/5): References**
> >
> > **REFERENCES:**
> > 1. EMaQ: Expected-Max Q-Learning Operator for Simple Yet Effective Offline and Online RL. Seyed Kamyar Seyed Ghasemipour, Dale Schuurmans, Shixiang Shane Gu
> > 2. DDCO: Discovery of Deep Continuous Options for Robot Learning from Demonstrations. Sanjay Krishnan, Roy Fox, Ion Stoica, Ken Goldberg
> > 3. Learning Robot Skills with Temporal Variational Inference. Tanmay Shankar, Abhinav Gupta
> > 4. Why Does Hierarchy (Sometimes) Work So Well in Reinforcement Learning? Ofir Nachum, Haoran Tang, Xingyu Lu, Shixiang Gu, Honglak Lee, Sergey Levine
> > 5. beta-VAE: Learning Basic Visual Concepts with a Constrained Variational Framework. Irina Higgins, Loic Matthey, Arka Pal, Christopher Burgess, Xavier Glorot, Matthew Botvinick, Shakir Mohamed, Alexander Lerchner
> > 5. Deep Reinforcement Learning with Double Q-learning. Hado van Hasselt, Arthur Guez, David Silver
> > 6. Soft Actor-Critic: Off-Policy Maximum Entropy Deep Reinforcement Learning with a Stochastic Actor. Tuomas Haarnoja, Aurick Zhou, Pieter Abbeel, Sergey Levine
> > 7. Auto-Encoding Variational Bayes. Diederik P Kingma, Max Welling
> > 8. Unsupervised Curricula for Visual Meta-Reinforcement Learning. Allan Jabri, Kyle Hsu, Benjamin Eysenbach, Abhishek Gupta, Sergey Levine, Chelsea Finn Neural Information Processing Systems (NeurIPS), 2019
> > 9. Dynamics-Aware Unsupervised Discovery of Skills. Archit Sharma, Shixiang Gu, Sergey Levine, Vikash Kumar, Karol Hausman
> > 10. Fixing a Broken ELBO. Alexander A. Alemi, Ben Poole, Ian Fischer, Joshua V. Dillon, Rif A. Saurous, Kevin Murphy
> > 11. Learning Latent Plans from Play. Corey Lynch, Mohi Khansari, Ted Xiao, Vikash Kumar, Jonathan Tompson, Sergey Levine, Pierre Sermanet
> > 12. Off-Policy Deep Reinforcement Learning without Exploration. Scott Fujimoto, David Meger, Doina Precup
> > 13. Near-Optimal Representation Learning for Hierarchical Reinforcement Learning. Ofir Nachum, Shixiang Gu, Honglak Lee, Sergey Levine
> > 14. Data-Efficient Hierarchical Reinforcement Learning. Ofir Nachum, Shixiang Gu, Honglak Lee, Sergey Levine

---

> > > ### Comment · AnonReviewer3 · 2020-11-20
> > > **Thank you for your very detailed response**
> > >
> > > Thank you for your very detailed response. This paper is filled with
> > > ideas, which makes it difficult to fully disentangle your contribution.
> > > Your reply and the updated draft considerably helps my understanding. I
> > > particularly appreciate the statements describing motivation in Section
> > > 1. The added theoretical discussion of Theorem 4.1 and 4.2 elucidates
> > > the main technical contribution and your reply clears up my
> > > misunderstandings regrading the optimality of the high-level controller
> > > in Theorem 4.2. I also acknowledge that there are indeed differences in
> > > the proof techniques between this work and Nachum et al. (2018).
> > >
> > > My only remaining concern is with the experiments. The results on
> > > Antmaze and Kitchen are certainly impressive. The added comparison to
> > > CARML and DADS also demonstrates the significant performance benefit of
> > > OPAL on Antmaze. However, I still think that the presented work is
> > > difficult to place in context. The binary nature of success in these
> > > environment undoubtedly exacerbates the performance difference. Do you
> > > have any thoughts on how these results may change if you report the
> > > cumulative return based on the "dense" reward signal in Antmaze (or
> > > other environments such as walker / hopper / halfcheetah and their
> > > offline datasets in D4RL)? The difference between OPAL and baseline will
> > > be smaller in this setting, but I imagine that temporally abstracted
> > > primitives will still help.
> > >
> > > Nachum, Ofir, Shixiang Gu, Honglak Lee, and Sergey Levine. 2018.
> > > “Near-Optimal Representation Learning for Hierarchical Reinforcement
> > > Learning.” *arXiv:1810.01257*. <http://arxiv.org/abs/1810.01257v2>.

---

> > > > ### Author Response · Authors · 2020-11-21
> > > > **Thank you for the follow up. Addressing experiment on evaluation with dense reward**
> > > >
> > > > We thank the reviewer for their prompt response. We are glad that our response helped elucidate the main technical contributions about the paper.
> > > >
> > > > To address the reviewer's remaining concern about the experiments, we have now evaluated the policies learned by OPAL, CARML, and DADS on two variants of dense reward **(Table 7)** -- **(i)** sum of negative L2 distances between the states and the target goal state in the  trajectory (equivalent to trajectory return under the negative state L2 distance reward) and **(ii)** cumulative negative L2 distances between the states in the last 5 time steps and the target goal (this measures the distance between the ant and the goal towards the end of the trajectory). We also plotted the heatmaps of the state visitation distribution for these different methods **(Figure 5)**. These are now added to Appendix F of the paper in purple. We additionally copy the table below.
> > > >
> > > > **Success rate**
> > > > CQL: 53.7 $\pm$ 6.1,
> > > > **CQL+OPAL: 81.1 $\pm$ 3.1**,
> > > > CQL+DADS: 59.1 $\pm$ 3.1,
> > > > CQL+CARML: 15.1 $\pm$ 2.6
> > > >
> > > > **Cumulative dense reward**
> > > > CQL: -12138.6 $\pm$ 720.3,
> > > > **CQL+OPAL: -7795.7 $\pm$ 535.4**,
> > > > CQL+DADS: -11184.7 $\pm$ 610.1,
> > > > CQL+CARML: -13387.3 $\pm$ 710
> > > >
> > > > **Cumulative dense reward (last 5 time steps)**
> > > > CQL: -45.1 $\pm$ 9.2,
> > > > **CQL+OPAL: -7.8 $\pm$ 4.6**,
> > > > CQL+DADS: -33.1 $\pm$ 8.1,
> > > > CQL+CARML: -51.6 $\pm$ 5.6
> > > >
> > > > These results indicate that CARML and DADS are worse in terms of both the dense reward metrics when compared with OPAL, meaning that they do not reach as close to the final goal as OPAL.  Visualizing the heatmap helps us reason about this poor performance of CARML and DADS. Note that the majority of the density of CARML and DADS is spent at “dead-end” corners in the maze, where the ant is prone to getting stuck, unlike OPAL which generally proceeds straight to the goal location.
> > > >
> > > > We hope that this response addresses the reviewer’s concern. Please let us know, and we will be happy to address additional concerns if any.

---

> > > > > ### Comment · AnonReviewer3 · 2020-11-25
> > > > > **I appreciate the additional experimental results**
> > > > >
> > > > > I appreciate the additional experimental results and the thorough reply. I will re-evaluate the paper in the light of this discussion, and the updated draft.

---

> ### Author Response · Authors · 2020-11-19
> **Author Response (Part 1/5): Comparison to other offline unsupervised skill discovery methods**
>
> We thank the reviewer for their constructive feedback. To address the reviewer’s concerns, we now provide a discussion of hyperparmeters in Appendix E and we also added a comparison to other skill discovery methods in Appendix F. We have also edited the motivation in Section 1 to clearly indicate that the main motivation of this work is to theoretically and empirically demonstrate the efficacy of primitive learning in offline RL. We answer the remaining reviewer’s questions as follows:
>
> **“It would be helpful to directly compare to other skill-discovery methods that have been applied to online RL but can be adapted to offline RL. For example, what prevents the work by Campos et al. (2020) and the baselines therein to be applied in conjunction with an offline RL algorithm?”**
>
> **RESPONSE:** Although we did not include it in the initial version of our paper, we had tried offline variants of existing unsupervised skill discovery algorithms and we now include them in Appendix F. These algorithms include CARML (Jabri et al, 2019) and DADS (Sharma et al, 2019) and we compare them to OPAL on the antmaze-medium (diverse) environment in Appendix F. While EDL (Campos et al.) can’t be directly used in offline settings as it requires online interaction to learn the primitive policy $\pi_{\theta}(a|s,z)$, its offline variant will be similar to the offline variant of CARML. In addition to the results, we have included a discussion on these alternative offline unsupervised primitive discovery methods in Appendix F of the paper.
> We will now summarize the findings here as well, which empirically are significantly worse than OPAL.
>
> Results on antmaze medium (diverse):
> - CQL-> 53.7 $\pm$ 6.1
> - CQL+Offline CARML-> 15.1 $\pm$ 2.6
> - CQL+Offline DADS> 59.1 $\pm$ 3.1
>
> To understand the possible reasons behind this trend in performance, we note that offline CARML treats the states in the trajectory conditionally independent of each other given the latent vector $z$. Therefore, it clusters states together which are spatially close to each other and hence is unable to separate out different control modes from the same starting locations. This degrades the learned primitives and explains the poor performance of CQL+Offline CARML given it learns the task policy on top of not so great primitives.
>
> On the other hand, offline DADS treats the state transitions in the trajectory conditionally independent of each other given $z$ and thus clusters trajectories with similar state transitions together. This allows it to more effectively separate out the different control behaviors in the data. Therefore, CQL+offline DADS slightly improves upon CQL but it is still limited by a discrete number of skills. Furthermore, increasing the number of skills from $10$ to $20$ doesn’t help and gives similar performance. Since both CARML and DADS use Bayes rule to calculate $p_{\phi,\omega}(z|\tau)$,  it is intractable to use continuous skill space Therefore, we switched to learning a $\beta$-VAE (higgins et al, 2016) style generative model with continuous skill space i.e. OPAL.
>
> We visualize the trajectory clustering obtained by Offline CARML and Offline DADS in Figure 4 of the paper (Appendix F). Table 6 in Appendix F contains an ablation study on the number of skills used for both these methods. Using the hyperparameters from Nachum et al, 2019, we fix the length of the sub trajectory $c=10$ for these experiments.

---

> > ### Author Response · Authors · 2020-11-19
> > **Author Response (Part 2/5): Hyperparameter selection**
> >
> > **“It doesn't help that there is no discussion of how hyperparameters were chosen for each component...”
> > “Section 5.1, results: An ablation study is discussed but I cannot find the corresponding results table/figure in the paper.”**
> >
> > **RESPONSE:** We have now added a discussion of the hyperparameters for OPAL in Appendix E. The main hyperparameters to choose for OPAL are hyperparameters for the $\beta$-VAE: (i) dimension of latent vector $z$, (ii) length of sub-trajectory $c$,  (iii) the architectural choices of encoder, prior and primitive policy, and (iv) coefficient $\beta$ used in the loss function. We will now summarize the rationale behind the choices we made (i)-(iv). Note that these choices were predominantly guided by previously established conventions, rather than extensive hyperparameter tuning on our part.
> >
> > (i)&(ii) **Value of z and c:** We included the ablation study on choice of $\mathrm{dim}(\mathcal{Z})$ and $c$ in Appendix E of the paper.  This is already briefly mentioned in Section 5.1 of the paper, but now, we have included a detailed discussion in Appendix E which describes our rationale behind our choices for $\mathrm{dim}(\mathcal{Z})$ and $c$. We also describe the discussion here:
> >
> > As shown in Table 5 (Appendix E) of the paper, we experimented with different choices of $\mathrm{dim}(\mathcal{Z})$ on antmaze-medium (diverse). Using the hyperparameters from Nachum et al. 2018, we fixed $c=10$. While $\mathrm{dim}(\mathcal{Z})$ gave similar performances (success rate for $\mathrm{dim}(\mathcal{Z})=8$ was 81.1 $\pm$ 3.1 and for $\mathrm{dim}(\mathcal{Z})=16$ was 81.3 $\pm$ 1.8), $\mathrm{dim}(\mathcal{Z})=4$ performed slightly worse (success rate of 68.7 $\pm$ 2.3). Therefore, we selected $\mathrm{dim}(\mathcal{Z})=8$ for our final model as it was simpler.
> >
> > **Temporal abstraction actually helps:** To empirically verify that the gain in performance was due to temporal abstraction and not better action space learned through latent space, we tried $c=1$ ($\mathrm{dim}(\mathcal{Z})=8$) and found the performance to be similar to that of CQL (i.e. $55.3\pm3.8$) thereby empirically supporting the theoretical benefits of temporal abstraction.
> >
> > We found $\mathrm{dim}(\mathcal{Z})=8$ and $c=10$ to work well with other environments as well. However, we acknowledge that the performance of CQL+OPAL can be further improved by carefully choosing better hyperparameters for each environment or by using other offline hyperparameter selection methods for offline RL, which is a subject of future work.
> >
> > (iii) **architectural choices:** Since the encoder processes a sequence of states and actions, we used a standard fully connected  network with 2 hidden layers (ReLU activation) to preprocess the states (in the sequence) and fed it to a standard bidirectional GRU with 4 hidden layers to process the sequence of actions and state features. Similarly, for the prior which only takes in a single state as input, we used a standard FC network with 2 hidden layers (ReLU activation). We modelled the latent vector $z$ as a gaussian distribution as it has been a simple, common and reasonably effective choice in past papers (Kingma et al, 2013, Higgins et al, 2016, Lynch et al 2019). Finally, we used a standard fully connected network with 2 hidden layers and ReLU activation for the antmaze medium and antmaze large (diverse) environments. However, for kitchen mixed and kitchen partial environments, we took cues from EMAQ (Ghasemipour et al, 2020) and used an autoregressive primitive policy with architecture exactly equal to that used in EMAQ.
> >
> > (iv) **Beta value:** Taking cues from past papers (Lynch et al 2019, Alemi et al 2017) which observed that setting $\beta \le 1$ was enough to prevent posterior collapse, we went with $\beta=0.1$ taken from Lynch et al (2019).

---

> > > ### Author Response · Authors · 2020-11-19
> > > **Author Response (Part 3/5): Online RL baseline discussion and Motivation of OPAL from Offline perspective**
> > >
> > > **“Section 5.2, Table 3: With so many 0.0's, this leads me to believe that the baselines are quite weak, or the problem is too hard for the baselines. How were the baselines chosen. For example, why was DDCO paired with DDQN instead of SAC?”**
> > >
> > > **RESPONSE:** The antmaze-medium and antmaze-large tasks are very hard tasks for any reinforcement learning algorithm. To solve this task through online RL, we need both (i) hierarchy (i.e. learning a policy on top of primitives) which improves exploration (Nachum et al, 2019) and (ii) unlabelled (i.e. no task reward) offline dataset which allows us to bootstrap the primitives. This informed our choice of baselines, which we summarize below. Note that these baselines encompass SOTA algorithms and use benchmarked open-source implementations whenever possible, ensuring fair comparison.
> > >
> > > (i) **Importance of unlabelled offline dataset allowing us to bootstrap our primitives:** We start with randomly initialized primitives and train both the high level policy and the primitive policy with HIRO (Nachum et al, 2018) using the official implementation from https://github.com/tensorflow/models/tree/master/research/efficient-hrl and observe that it achieves a low success rate with both sparse and dense rewards on the antmaze domains.
> > >
> > > (ii) **Importance of hierarchy which helps with exploration:** We pretrain a flat policy with offline (unlabelled) dataset and finetune it with Soft Actor Critic (Harnooja et al, 2018) using the rlkit codebase (https://github.com/vitchyr/rlkit) with standard hyperparameters. It achieves 0 success rate with both sparse and dense rewards on the antmaze domains. This is because a flat policy is unable to capture the multi-modality of the dataset. However, learning a latent conditioned primitive policy (using beta VAE, i.e. OPAL) is able to capture the multi-modality in the dataset and then performing online RL on top of learned skills allows us to explore better and significantly improve the performance in all the tasks.
> > >
> > > (iii) **Comparison to DDCO (Krishnan et al, 2017):** We based our implementation on the open sourced codebase: https://github.com/royf/ddo to extract discrete set of skills (20 skills) as DDCO extracts a discrete set of continuous options. We learned a policy on top of these discrete skills with DDQN (Hasselt et al, 2015) using the rlkit codebase (https://github.com/vitchyr/rlkit) with standard hyperparameters. We didn’t use SAC since the standard implementation of SAC assumes continuous action space and DDCO is necessarily a discrete-primitive algorithm. However, the discrete skills extracted with DDCO turned out to be limiting and hence it also didn’t perform well in the task.
> > >
> > > The above points are also discussed briefly in Appendix A.2 of the paper.
> > >
> > > **“The proposed method is not specifically designed to leverage anything particular in offline RL. As you show, it can be applied to online RL. However, why is this motivated from the offline perspective?”**
> > >
> > > **RESPONSE:** While our method can also be applied to online RL problems as we show, our main motivation is to use effective (and simple) ways to accelerate offline RL. We have now made this very explicit in Section 1 of the paper.  One such effective way is to first extract temporally extended primitives from offline data and then leverage those primitives to accelerate offline RL algorithms. While this might seem intuitive at first, it is actually not so intuitive, given the fact that prior works that performed controlled empirical analysis on the benefits of hierarchical RL (Nachum et al, 2019) find that hierarchies work better than standard online RL because of improved exploration and there is absolutely no exploration to be done in an offline RL setting. To the best of our knowledge, ours is the first work to theoretically justify (Theorem 4.1 and 4.2) and experimentally verify the benefits of hierarchy in offline settings, showing that hierarchies can provide temporal abstraction that allows us to reduce the effect of compounding errors issue in offline RL. We also added an experiment (Appendix E) showing that indeed hierarchy helps offline RL via temporal abstraction. This is the reason behind our motivation towards demonstrating the efficacy of learning primitives in an offline RL perspective. We also reply to this question in more detail in a shared response across all reviewers.

---

### Author Response · Authors · 2020-11-19
**Shared response to Reviewers and AC (Part 1/2): Major revision changes to the paper**

**SUMMARY OF UPDATES/REVISIONS**

We thank all the reviewers for their feedback. We have uploaded the updated paper which incorporates many of the suggestions of the reviewers. To aid the reviewers, we have marked all the changes in blue. We have made the following changes:

1) We have omitted the success rate of CQL+OPAL (on antmaze environments) from the description of Figure 1 to prevent confusion -- this result is already included in the experiments section.

2) The diagram of OPAL overview (Figure 2) has been updated. The new diagram tries to better highlight the training procedure of OPAL, its different components and how it is used during test time. For additional visualizations, we added Figure 3 which shows state visitation heatmaps for the learned offline policies.

3) We have fixed minor text issues as highlighted by the reviewers and improved the text in general by fixing minor issues.

4) We updated the results of CQL+OPAL on kitchen mixed and kitchen partial environments. Taking cues from EMAQ (Ghasemipour et al, 2020), we discretized the action space and used an autoregressive model to represent the primitive policy for kitchen tasks. We borrowed the discretization scheme and the autoregressive model from EMAQ. With these changes, we outperform EMAQ on kitchen partial and match its performance on kitchen mixed as shown in Table 1 of the paper.

5) We have improved the clarity of the statement of Theorem 4.2 in the main text. However, the rigorous version of the theorem along with the proof  (Appendix B) remains unchanged.

6) We have fixed the text below Corollary 4.1.1 to say that $\epsilon_c$ would decrease as a result of optimization.

7) We added Appendix E which contains ablation studies on how we chose dimension of latent variable $z$ and the length of the sub-trajectory $c$.

8) We added Appendix F which includes a thorough discussion and experimental results on other unsupervised skill extraction techniques (Offline CARML and Offline DADS) we had tried in the early phases of the project and how they don’t perform as well as OPAL. It also visualizes the trajectory clustering obtained by Offline CARML and Offline DADS.

---

> ### Author Response · Authors · 2020-11-19
> **Shared response to Reviewers and AC (Part 2/2): Clarification on Motivation and Contributions**
>
> **ADDRESSING CONCERNS REGARDING MOTIVATION/CONTRIBUTIONS**
>
> One concern expressed by the reviewers was that the motivation of the paper was vague and it was unclear why we motivated our paper from the perspective of offline RL when it can be also used for online RL. We would like to answer these questions here:
>
> **Main Contribution:** We would like to state that we believe our contribution is not to come up with the best possible offline skill discovery method. We are trying to show even using a simple offline skill extraction method with $beta$-VAE inspired generative model (i.e. OPAL) can be effective and significantly help with performance of Offline RL as shown theoretically and demonstrated empirically in our paper. To the best of our knowledge, ours is the first work  to theoretically justify and experimentally verify the benefits of primitive learning in *offline* settings, and that’s why we motivate our method from an offline perspective. We have now updated the motivation in our paper to reflect this.
>
> **Motivation:** The main motivation of the paper is to come up with an effective (yet simple) way to improve offline RL. Past works (Krishnan et al. 2017, Shankar et al. 2020) have shown that learning with skills (learned from data) accelerates online RL and the main reason behind this is improved exploration, as suggested by Nachum et al. 2019. However, it isn’t clear if having a hierarchy would help in offline RL, given that the offline dataset is fixed and there’s no exploration to be done. Our initial intuition was that hierarchy could still help in offline RL if the learned temporally extended skills are good. This is because it will effectively reduce the task horizon and make the RL optimization easier by reducing the curse of error propagation. We tested different ways (OPAL and alternative methods as described in Appendix F of the paper) to extract temporally extended skills from the offline dataset. We found that extracting a continuous set of skills with a $\beta$-VAE (higgins et al 2016) inspired generative model style generative model (i.e. OPAL) and then learning a task policy on top of the primitives with offline RL (CQL) significantly improved our performance of naive offline RL methods.
>
> We also  mathematically formalize our intuition of (1) why learning good primitives are important and (2) how offline RL (CQL) helps in the optimization once the primitives learned are good? We answered (1) using Theorem 4.1. The essential crux of Theorem 4.1 is that if “good” primitives are learned, then the suboptimality of the policy learned over the primitives will be small when compared to the optimal policy for a task.  Formally, Theorem 4.1 relates this suboptimality of primitives to the final autoencoding loss, dimension of the primitives and length of the temporally extended primitives ($c$). We answer (2) with Theorem 4.2 which shows how temporally extended primitives allow us to gain asymptotically in terms of performance (Theorem 4.2) due to better credit assignment resulting from reduced effective task horizon. Our analysis holds for any “conservative” offline RL algorithm that encourages the learned offline policy to be close to the policy generating the dataset. . In practice, this bound implies an improvement in performance, as it is easier to control the suboptimality in skill discovery and this is verified in our experiments.
>
> REFERENCES:
> 1. EMaQ: Expected-Max Q-Learning Operator for Simple Yet Effective Offline and Online RL. Seyed Kamyar Seyed Ghasemipour, Dale Schuurmans, Shixiang Shane Gu
> 2. DDCO: Discovery of Deep Continuous Options for Robot Learning from Demonstrations. Sanjay Krishnan, Roy Fox, Ion Stoica, Ken Goldberg
> 3. Learning Robot Skills with Temporal Variational Inference. Tanmay Shankar, Abhinav Gupta
> 4. Why Does Hierarchy (Sometimes) Work So Well in Reinforcement Learning? Ofir Nachum, Haoran Tang, Xingyu Lu, Shixiang Gu, Honglak Lee, Sergey Levine
> 5. beta-VAE: Learning Basic Visual Concepts with a Constrained Variational Framework. Irina Higgins, Loic Matthey, Arka Pal, Christopher Burgess, Xavier Glorot, Matthew Botvinick, Shakir Mohamed, Alexander Lerchner

---

### Decision · Program_Chairs · 2021-01-07
**Final Decision**

**Decision:**

Accept (Poster)

**Comment:**

This paper presents an interesting mix of new theoretical and empirical results showing how learning temporally extended primitive behaviors can help improve offline (batch) RL.

Although 2/3 reviewers initially raised concerns regarding the motivation of the approach and some of the choices that were made, the authors did an excellent job at addressing these concerns in detail, and there is now a consensus towards acceptance.

I consider that this work is a meaningful contribution towards better offline RL, which is definitely a very important use case in practice. The authors have given convincing explanations to motivate their approach, and made several improvements to the paper. As a result, I am recommending it for acceptance, as a poster.